# ONLINE CONTINUAL GRAPH LEARNING

## ABSTRACT

The aim of Continual Learning (CL) is to learn new tasks incrementally while avoiding catastrophic forgetting. Online Continual Learning (OCL) specifically focuses on learning efficiently from a continuous stream of data with shifting distribution. While recent studies explore Continual Learning on graphs exploiting Graph Neural Networks (GNNs), only few of them focus on a streaming setting. Many real-world graphs evolve over time and timely (online) predictions could be required. However, current approaches are not well aligned with the standard OCL literature, partly due to the lack of a clear definition of online Continual Learning on graphs. In this work, we propose a general formulation for online Continual Learning on graphs, emphasizing the efficiency of batch processing while accounting for graph topology, providing a grounded setting to analyze different methods. We present a set of benchmark datasets for online continual graph learning, together with the results of several methods in CL literature, adapted to our setting. Additionally, we address the challenge of GNN memory usage, as considering multiple hops of neighborhood aggregation can require access to the entire growing graph, resulting in prohibitive costs for the setting. We thus propose solutions to maintain bounded complexity for efficient online learning.

## 1 INTRODUCTION

In traditional machine learning, models are at first trained in an environment where all training data is simultaneously available to the learning algorithm, and only at a later time model predictions are produced on new input data. Importantly, data observations are assumed to be mutually independent and identically distributed. Real-world environments however often generate data in chunks or streams which often entail shifts in the data distribution or even variations in tasks to be solved. In turn, previously trained models may require expensive retraining or model reconfiguration to stay up to date. In this setting, Continual Learning (CL) (Parisi et al., 2019; De Lange et al., 2022), lifelong learning (Chen & Liu, 2018) and incremental learning (Chaudhry et al., 2018a) are similar machine learning paradigms sharing the same goal of adapting models to incrementally learn as soon as new data and new tasks are presented.

Even further, in the *online* learning setting, training data points are collected sequentially, processed in real-time by the learning method, and immediately discarded (Chaudhry et al., 2018b; Mai et al., 2022). Such strict environments are found in monitoring and control problems (Zliobaite et al., 2016; Gunasekara et al., 2023), such as traffic management, activity recognition, fraud detection, with other applications such as on data generated by optical sensors (Souza et al., 2020) or prediction of power production considering environmental conditions (Lobo et al., 2020). Online CL is therefore an extremely challenging setting that requests models to quickly adapt to new conditions, allowing for anytime inference with minimal latency and without forgetting already acquired knowledge.

Research on CL extends to graph-structured data as well, often referred to as Continual Graph Learning (CGL) (Yuan et al., 2023). Indeed, many machine learning tasks naturally represent data as graphs and address predictions of properties of single nodes or edges, or entire graphs. Common examples include social networks, citation networks, and biological systems, where relationships between entities are modeled as graphs. However, most graphs in the real world are not static: they continuously evolve over time experiencing the addition/removal of nodes and modifications to the topology. Examples are seen in dynamic environments such as social networks, citation networks, and traffic systems, where new users join, papers are published, and road conditions may change (Liu et al., 2021; Zhou & Cao, 2021). Countless deep learning models for graph-

structured data, such as Graph Neural Networks (GNNs), make predictions by relying on node-level representation computed from neighboring nodes. While this enables rich representations that condition the predictions on related observations, it poses unique challenges associated with the online setting, due to the need for neighboring, past information to make predictions for new nodes, as well as memory and computational issues due to the rapid growth of certain graph families.

In this paper, we introduce three main contributions to address these challenges. **(1)** We formalize the Online Continual Graph Learning (OCGL) framework, establishing a foundation for Continual Learning on graphs in a node-level streaming environment. By doing so, we bridge the gap between Online CL and CGL settings. **(2)** We present a benchmarking environment for OCGL. We introduce four benchmark datasets and conduct a first set of extended experiments to evaluate various CL methods, that we suitably adapted to operate under the OCGL setting. We use a hyper-parameter selection protocol tailored for online learning that ensures a fair comparison between CL techniques. Insightful conclusions are drawn from the analyses, highlighting the higher performance of replay-based methods, especially A-GEM. **(3)** We bring the attention of the research community to the issue of neighborhood expansion which could undermine the online computational efficiency due to the complexity of multi-hop aggregation in GNNs. To address it, we propose a first, simple solution to ensure that models can efficiently scale as the graph evolves. We conduct a second set of experiments with the proposed neighborhood sampling solution. This work leaves room for further research to develop more effective approaches to tackle the problem.

## 2 BACKGROUND AND RELATED WORKS

**Continual Learning.** Depending on the type of shift in the data distribution, CL has been categorized into three main scenarios (Van De Ven et al., 2022): in *task-incremental* learning, the model sequentially learns distinct tasks, which requires availability of task identifiers to make predictions; *class-incremental* learning consists in classifying instances with an increasing number of classes, without task identifiers; finally, *domain-incremental* learning requires solving the same problem in different contexts. In the past, the main applications of CL included reinforcement learning (Kirkpatrick et al., 2017; Rolnick et al., 2019) and especially computer vision (Rebuffi et al., 2017; Lopez-Paz & Ranzato, 2017; Aljundi et al., 2018; Li & Hoiem, 2018; Masana et al., 2022; Mai et al., 2022), even though most of the methods that have been developed to address these problem domains are not limited to these fields and can be used for a wide range of other machine learning tasks. Three main broad categories of CL approaches to mitigate forgetting have been proposed in the literature (De Lange et al., 2022): *regularization* methods, *replay* methods and *architectural* methods. Regularization methods (Kirkpatrick et al., 2017; Zenke et al., 2017; Aljundi et al., 2018; Li & Hoiem, 2018; Chaudhry et al., 2018a) introduce additional loss terms to preserve important parameters to retain previously acquired knowledge. Replay methods (Rebuffi et al., 2017; Lopez-Paz & Ranzato, 2017; Chaudhry et al., 2019; 2018b) use a memory buffer to store some representative samples from old tasks, to use them jointly with new samples to update model parameters. Architectural methods (Fernando et al., 2017; Masse et al., 2018; Rusu et al., 2022) avoid changes to model parameters either by gating mechanisms or by introducing new parameters, allowing the model to grow in size.

**Online Continual Learning.** In the usual Continual Learning scenarios described above, data arrive one task at a time, allowing for *offline* training with multiple passes over the data for the current task (De Lange et al., 2022). A more realistic yet challenging scenario is that of *Online Continual Learning (OCL)* (Chaudhry et al., 2018b; Mai et al., 2022; Soutif–Cormerais et al., 2023), where data arrive in small batches of only few samples, without the possibility for the model to store all the data for the current task, either for privacy reasons or memory limitations. In this setting the learning algorithm must be able to use efficiently the mini-batches that arrive in the non-stationary stream. Additionally, whereas for CL we assume to know the task boundaries, OCL can be performed in a boundary-agnostic setting, or task-free, potentially allowing for more varied distribution shifts compared to the three settings described before (Koh et al., 2021). Many CL methods though are not suited to this setting and require modifications. An additional characteristic of OCL is the ability to perform anytime inference: the model should always be up-to-date and ready to make predictions online after each training batch, reacting quickly to distribution shifts (Koh et al., 2021).

**Learning on graphs.** *Graph Neural Networks (GNN)* (Sperduti & Starita, 1997; Scarselli et al., 2009; Micheli, 2009; Kipf & Welling, 2017) have emerged as the state-of-the-art approach for

dealing with network data, generalizing convolution to graph structures. The core mechanism of most GNNs is message passing (Gilmer et al., 2017): at each layer, the hidden embedding $h_v^{(k)}$ of each node $v$ is updated using information from its neighborhood $\mathcal{N}(v)$ as $h_v^{(l+1)} = \mathrm{UPDATE}(h_v^{(l)}, \mathrm{AGGREGATE}(\{h_u^{(l)} : u \in \mathcal{N}(v)\}))$. Here AGGREGATE and UPDATE are differentiable functions specified by the particular model. Specifically, as at each step each node updates its embedding using the information (message) coming from its neighbors, after $l$ layers it will depend on its $l$-hop neighborhood. Graph-based processing of temporal data has a relatively short history, primarily encompassing the study of temporal graphs (Kazemi et al., 2020; Gravina & Bacciu, 2024) and time series data (Cini et al., 2023a; Jin et al., 2024) with dedicated adaptation strategies to deal with evolving graphs (Cini et al., 2023b) and benchmarks (Huang et al., 2023).

**Continual Graph Learning.** In the past few years researcher started to develop CL strategies tailored to graph data (Wang et al., 2020), with applications such as recommender systems (Xu et al., 2020) and traffic prediction (Chen et al., 2021). The main approaches resemble the general CL strategies, with a particular focus on preserving topological information with a loss term on neighborhood aggregation parameters (Liu et al., 2021), or specific node selection policies to retain informative nodes as memory buffer (Zhou & Cao, 2021). Recently a number of surveys have been published on the topic (Febrinanto et al., 2023; Yuan et al., 2023; Zhang et al., 2024; Tian et al., 2024), and a benchmark has been proposed (Zhang et al., 2022). *Continual Graph Learning (CGL)* possesses some peculiarities that differentiate it from other problem domains with independent samples, as graph structure requires careful consideration. Specifically, we can distinguish between *graph-level* CGL and *node-level* CGL (Zhang et al., 2022). In the first, the model needs to make predictions about entire graphs, and thus each sample in the training data is an independent graph. Standard Continual Learning settings and methods can thus be applied with relative ease (Carta et al., 2022). Instead, in node-level GCL predictions are made about the single nodes in usually one single graph. In this node-leve CGL setting each new task therefore consists a new subgraph of the overall graph, with new classes or type of nodes. Specifically, the task subgraph arrives all at once, and offline training is performed on it. Here, an additional specification needs to be made about the availability of inter-task edges (Tian et al., 2024): since message passing GNNs aggregate information from neighboring nodes, the presence of inter-task edges towards nodes of previous tasks would inevitably lead to the use of past data. In practice, inter-task edges are often kept, but without access to the labels from past tasks (Zhang et al., 2024). Finally, the addition of new edges creates an additional source of backward interference: when evaluating the model on nodes belonging to past tasks, their neighborhood composition and topology will be changed compared to when the task was observed.

The Online Continual Learning setting described before has been explored in domains like computer vision (Mai et al., 2022; Soutif–Cormerais et al., 2023) and on sequences (Parisi & Lomonaco, 2020), but to the best of our knowledge it has not yet been applied to graphs. Some papers on CGL consider a setting referred to as *streaming* (Wang et al., 2020; Perini et al., 2022), yet the approaches can be categorized as offline CL as the streams consist of graph snapshots (often corresponding to entire tasks or nonetheless not classifiable as mini-batches), on which models are trained with multiple passes.

## 3 ONLINE CONTINUAL GRAPH LEARNING

This section introduces *Online Continual Graph Learning (OCGL)*, a framework that ports CGL to the online problem setting. OCGL is particularly applicable to dynamic real-world scenarios such as social networks or recommender systems, where sudden distribution changes occur, and quick model adjustments are essential for making anytime predictions.

### 3.1 PROBLEM FORMULATION

**A growing network.** We model the data associated with an OCGL problem as an evolving graph $\mathcal{G}$ induced by a stream of nodes $v_1, v_2, \ldots, v_t, \ldots \in \mathbb{N}$. At every time-step $t$, the monitored system is represented as a graph $\mathcal{G}^t = (\mathbb{V}^t, \mathbb{E}^t, \boldsymbol{X}^t)$ defined by node set $\mathbb{V}^t = \{v_1, \ldots, v_t\}$, edge set $\mathbb{E}^t \subseteq \mathbb{V}^t \times \mathbb{V}^t$, and a set of node attributes $\boldsymbol{X}^t = \{\boldsymbol{x}^i\}_{i \leq t} \subset \mathbb{R}^F$. Edge attributes, e.g., accounting for edge directions or defining the type of node-node relations, can be considered likewise, however, they are here excluded to ease the presentation. The graph nodes $v_i$ can be associated with

class labels $y_i \in \{1, \dots, C\}$ collected in set $Y^t = \{y_i\}_{i \leq t}$ to be predicted and/or used as training samples to learn the model. Graph $\mathcal{G}^t$ is a snapshot of the temporal graph $\mathcal{G}$ which, together with $Y^t$, collects all information available up to time step $t$. At every time step $t$ a new node is acquired from the monitored system and populates the current snapshot graph $\mathcal{G}^{t-1}$. Specifically, tuple $(v_t, \mathcal{N}(v_t), \boldsymbol{x}^t)$ containing a new node index $v_t \notin \mathbb{V}^{t-1}$, associated node features $\boldsymbol{x}_t$, and a set of neighbors $\mathcal{N}(v_t) \subseteq \mathbb{V}^{t-1}$ is presented and connected to graph $\mathcal{G}^{t-1}$ according to the relations contained in $\mathcal{N}(v_t)$. Finally, target class label $y_t$ of node $v_t$ may or may not be acquired contextually to $(v_t, \mathcal{N}(v_t), \boldsymbol{x}^t)$; for instance, a prediction for node $v_t$ might be requested at time $t$ while the true class label $y_t$ is observed only at a later time.

**Problem statement.** The goal of OCGL is to learn a model $f_\theta$ to predict class label $y_t$ only from the subgraph of $\mathcal{G}_t$ associated with $v_t$ and its neighbors of order 1 or more. Parameters $\theta$ have a predefined (maximum) dimension and model $f_\theta$ is trained incrementally as soon as new nodes and the associated labels are provided. New nodes are acquired either individually or in small minibatches, slightly weakening the online setting as commonly done in the literature (Chaudhry et al., 2018b). Moreover, minibatches are seen only once and, after prediction and/or training is performed the mini-batch is discarded. As per the CGL problem setting, we allow the task to change over time. This requirement on small mini-batches is set to meet the need to perform anytime predictions Koh et al. (2021) where dealing with the entire graph $\mathcal{G}^t$ – or a large subgraph of it – is unfeasible either memory-wise or computationally. Specifically, we require the training on each mini-batch to have bounded compute and memory budgets. Although the size of the node mini-batch may vary between applications, we assume it to be small enough that using only edges within the mini-batch would not provide sufficient context for effective learning. In contrast to some CGL settings where training can be done on a task-specific subgraph, in OCGL the mini-batch is too small to be treated as a meaningful graph on its own, requiring to access neighborhood information.

**Mini-batching.** Within the introduced OCGL framework, and depending on the application, mini-batches can include L-hop neighbors with $L \geq 1$. Considering $L > 1$ does not conflict with the growing nature of the graph. Instead, it reflects the requirements of the predictive model $f_\theta$, which may rely on aggregating multi-hop information. As such, to construct L-hop neighborhoods, an up-to-date snapshot $\mathcal{G}^t$ is assumed to be stored in a *Past Information Store (PIS)* system, separate from an eventual memory buffer associated with predictive model $f_\theta$, as in the more general lifelong-learning setup (Chen & Liu, 2018). We do not impose memory limitations on the PIS to allow graph growth, but we require the training on each mini-batch to have bounded computational time and memory cost. That is, we assume to make limited use of information from the PIS for each mini-batch, and to only have access to the labels of nodes in the current batch. The definition of the evolving graph is general as it does not make assumptions on the distribution shifts happening in the node stream. This general, task free setting can be easily adapted or made more specific depending on the node stream: while a real-world stream could be induced by a time-stamp on the nodes, this setting can be derived from any static graph by establishing an ordering on the nodes. The three CL scenarios of task-, class- and domain-incremental can thus be easily adapted to an online setting by ordering nodes by task, similarly to what is done in other domains (Mai et al., 2022; Soutif–Cormerais et al., 2023).

## 3.2 Neighborhood expansion problem

The efficiency requirement for online learning poses non-negligible issues associated with reiterated message passing within multi-layer GNNs. As commented above, at each layer of the $L$ layer, the GNN aggregates the embeddings of the neighboring nodes, thus requiring access to $L$-hop neighborhoods. The size of the $L$-hop neighborhood however scales as $O(d^L)$ where $d$ is the average degree. Moreover, $d$ is time-dependent and can increase as the graph grows. Depending on how well-connected a graph is, very few hops may be required to go from any node to almost any other one; empirical evidence on the neighborhood growth is depicted in Figure 2 and those in Appendix E. Thus, having high $d$ or $L$ would require processing a number of nodes in the order of the entire (growing) graph for each mini-batch, going against efficiency in our definition of OCGL. It is therefore of paramount importance to keep a low $L$ or to introduce a strategy to mitigate $d$.

In cases where the topology of the full graph or the maximum node degree are known a-priori (such as in citation networks, where we expect the number of references for an article not to explode even if the body of literature increases), fixing a low number of layers can be a good solution. This is

the first strategy we evaluate in Section 6. Yet, in many real applications we may not have prior knowledge on the evolution of average degree. In such cases, even choosing a model with a low number of layers may be problematic, as the average degree could grow indefinitely and with it the computational complexity and memory usage for batch training. This issue is similar to the problem of scaling static GNNs for large graphs, for which mini-batch training is required both for memory and efficiency reasons. Numerous approaches have been developed to address this problem, such as fixing a number of neighborhood sampled for aggregation (Hamilton et al., 2017; Chen et al., 2018) or training on partitions of the graph (Chiang et al., 2019). In our context, the simplest solution seems to fix the number of neighbors for aggregation through sampling, which guarantees an upper bound on the size of the computational graph for each batch. Results with neighborhood sampling are reported in Section 7, after an empirical assessment of the problem of neighborhood expansion.

# 4 METHODS

Having defined the Online Continual Graph Learning setting, we consider and evaluate some popular CL techniques, most of which are agnostic with respect to the type of the input data. Some CGL learning strategies are not applicable to the online setting, such as ER-GNN (Zhou & Cao, 2021), which stores representative nodes according to metrics computed offline on an entire graph snapshot, and thus we resorted to a simplified version which performs reservoir sampling. Additionally, baselines that require expensive fine-tuning steps such as GDumb (Prabhu et al., 2020) are excluded, as it would violate the online setting. Overall, most strategies natively require task boundaries, and have been modified for the task free setting as described below.

1. **ER.** Experience replay (Chaudhry et al., 2019) is a simple yet powerful replay-based method, which selects samples to be stored in a memory buffer by reservoir sampling (Vitter, 1985). New incoming batches for training are then augmented with nodes sampled uniformly from the buffer.
2. **EWC.** Elastic Weight Consolidation (Kirkpatrick et al., 2017) adds a quadratic term to the loss to penalize the modification of important parameters. Parameter importance is approximated by the diagonal of the Fisher information matrix, which needs to be computed offline for each task. We therefore modify the algorithm to keep one single Fisher information matrix updated with a running average over the batches, similarly to the MAS approach detailed later. Another approach would be to keep a moving average, as done in EWC++ (Chaudhry et al., 2018a).
3. **A-GEM.** Averaged GEM (Chaudhry et al., 2018b) is a more efficient version of GEM (Lopez-Paz & Ranzato, 2017), which ensures that the average loss for past tasks does not increase. It achieves this by projecting the gradient of the incoming batch in the orthogonal space of the gradient computed on samples from a memory buffer, if their scalar product is negative. We select nodes for the buffer with reservoir sampling (Vitter, 1985).
4. **LwF.** Learning without Forgetting (Li & Hoiem, 2018) uses distillation (Hinton et al., 2015) to regularize the loss with logits from a previous version of the model (teacher) on the current batch. To use it in a task free setting, we introduce an additional hyperparameter: the number of batches after which the teacher is updated with the current model.
5. **MAS.** Memory Aware Synapses (Aljundi et al., 2018) is a quadratic regularization similar to EWC, but it calculates importance as the sensitivity of the output on parameters. MAS is natively an online method, as the importance scores are accumulated with each new data point.
6. **TWP.** Topology-aware Weight Preserving (Liu et al., 2021) is another regularization method, which preserves important weights for topological aggregation in GAT (Veličković et al., 2018), generalized also to other GNNs. We modify it for the online setting as EWC.

# 5 EXPERIMENTAL SETUP

In this section we introduce the specific experimental setup used, describing the construction of the node streams from benchmark datasets, and the details of model training and evaluation[1].

**Benchmarks.** Four node classification graph datasets are used in our experiments: CoraFull (Bojchevski & Günnemann, 2018), Arxiv (Hu et al., 2021), Reddit (Hamilton et al., 2017) and Amazon Computer (Shchur et al., 2019). The datasets are described in Appendix A. In order to position our experiments close to the rest of the Continual Learning literature, we devise a node stream derived

---

[1]Code available at: (supplementary material, link to be added after double-blind review)

from the class-incremental CL setting, which is considered as the most challenging one for catastrophic forgetting (Masana et al., 2022). We divide the nodes in the graph into groups with fixed order consisting of 2 classes: this would be the sequence of tasks in class-incremental learning (resulting in 35 tasks for CoraFull, 20 tasks for Arxiv and Reddit, and 5 for Amazon Computer), with task boundaries between pairs of classes. Then, we fix and ordering on the nodes of each task, and we stream the nodes accordingly. Therefore, the graph will gradually grow with mini-batches of nodes from two new classes at a time, which are processed in an online fashion. This allows us to consider metrics from the CL literature which require task boundaries, even though in our experiments the learning algorithm itself is task agnostic and simply adds a new output neuron when an instance of a new class is observed. For each dataset, we split the graph into 60% for training, 20% for validation and 20% for testing. A transductive setting is used: validation and test nodes are not used for loss computation, but they are still used for message passing.

**Performance assessment.** We consider three metrics widely adopted in the literature: *Average Accuracy (AA)*, *Average Forgetting (AF)* (Lopez-Paz & Ranzato, 2017), and *Average Anytime Accuracy (AAA)* (Caccia et al., 2021). AAA is obtained by evaluating the model on the validation set after each training batch, which we refer to as anytime evaluation. More details are reported in Appendix B.

**Training details.** In our experiments the backbone for all the Continual Learning strategies is the *Graph Convolutional Network (GCN)* (Kipf & Welling, 2017). For CoraFull, Arxiv and Amazon Computer datasets we use a 2-layer GCN with a fixed hidden dimension of 256 units as done by Zhang et al. (2022). On the Reddit dataset a single layer of GCN was used due to requirement of efficiency for OCGL (Section 3): with an average degree of 984, considering even two layers would require to use almost the entire graph for each mini-batch (this is further discussed in Section 7, where we provide results with neighborhood sampling instead). We use Adam optimizer (Kingma & Ba, 2017) without weight decay, tuning the learning rate as an hyperparameter with the protocol defined below. We consider the batch size to be fixed, as it could be dependent on the real world problem, and we experiment with two different sizes for each dataset. For the smaller datasets CoraFull and Amazon Computer we consider batches of 10 and 50 nodes, while for the larger Arxiv and Reddit we use sizes 50 and 250 simply for computational reasons. As suggested by Aljundi et al. (2019), multiple passes on the same mini-batch before passing to the next can be beneficial. We therefore considered as an additional tuned hyperparameter whether to perform multiple passes (5) on each batch. As a baseline, we use a *bare* model which is simply fine-tuned on the incoming stream without any CL strategy applied. Additionally, we provide an upper bound in the form of a model that is jointly trained offline on the entire graph.

**Hyperparameter selection.** Many works in the Continual Learning literature use a learning protocol that is akin to the classic machine learning setting, selecting hyperparameters by performing as many full passes over the task sequence as required by a grid search. This protocol violates stricter definitions of Lifelong Learning, where the stream is observed only once, and is indeed unrealistic for real applications where a model needs to quickly adapt to changes in data distribution. Chaudhry et al. (2018b) therefore proposed a more sensible hyperparameter selection protocol, which has now been used in several works (Xu et al., 2020; Mai et al., 2022; Soutif–Cormerais et al., 2023) and that we use for our experiments. With this protocol, only the first few tasks are used for hyperparameter selection, allowing the model to perform multiple passes, with the same online setting, over them to select the hyperparameters that lead to the best performance on validation nodes. In our case, due to the different number of tasks in the datasets used, we considered 20% of the tasks for this validation, with the exception of Amazon Computer where it was set to 2 as there are only 5 total tasks. Hyperparameters are then selected based on the AA on the validation set at this validation boundary. Details on the search space for the various methods are reported in Appendix C.

# 6 RESULTS ON FULL-NEIGHBORHOOD MINI-BATCHING

We start by discussing empirical results obtained on mini-batches containing the full neighborhoods of newly presented nodes, in contrast with next Section 7 where neighborhood sampling is analyzed. Although the neighboring expansion problem (Section 3.2) is present, the results of this section provide reference performance to be compared against the more realistic setting studied in the next section. Moreover, we discuss forgetting issues, the impact of the batch size, and the sensitivity to hyperparameters setting aside potential biases introduced by the neighborhood sampling. A com-

Table 1: Performance comparison on CoraFull with full neighborhood.

| METHOD | BATCH SIZE 10 | | | BATCH SIZE 50 | | |
|--------|---------------|--|--|---------------|--|--|
| | AA% ↑ | $\text{AAA}_{val}$% ↑ | AF% ↑ | AA% ↑ | $\text{AAA}_{val}$% ↑ | AF% ↑ |
| BARE | $15.97_{\pm0.99}$ | $24.52_{\pm0.87}$ | $-41.84_{\pm1.89}$ | $11.86_{\pm3.11}$ | $24.88_{\pm0.67}$ | $-77.67_{\pm3.37}$ |
| ER | $28.32_{\pm3.20}$ | $35.33_{\pm0.65}$ | $-63.29_{\pm3.59}$ | $13.74_{\pm2.04}$ | $30.00_{\pm0.21}$ | $-75.10_{\pm2.62}$ |
| EWC | $29.34_{\pm2.82}$ | $43.96_{\pm0.78}$ | $-20.06_{\pm5.40}$ | $29.29_{\pm1.45}$ | $46.55_{\pm2.31}$ | $-17.56_{\pm4.28}$ |
| A-GEM | $31.25_{\pm1.58}$ | $44.30_{\pm1.95}$ | $-57.88_{\pm2.45}$ | $30.22_{\pm2.25}$ | $39.97_{\pm0.64}$ | $-55.56_{\pm3.31}$ |
| LwF | $19.88_{\pm2.43}$ | $33.37_{\pm1.34}$ | $-48.64_{\pm3.53}$ | $20.70_{\pm1.87}$ | $28.28_{\pm0.66}$ | $-55.87_{\pm2.84}$ |
| MAS | $29.56_{\pm1.58}$ | $44.46_{\pm1.28}$ | $-15.35_{\pm1.69}$ | $30.08_{\pm1.00}$ | $52.01_{\pm1.30}$ | $-10.62_{\pm1.33}$ |
| TWP | $20.33_{\pm2.36}$ | $28.70_{\pm0.64}$ | $-64.81_{\pm2.22}$ | $26.61_{\pm2.65}$ | $36.45_{\pm1.42}$ | $-60.79_{\pm3.14}$ |
| JOINT | $67.55_{\pm0.05}$ | - | - | $67.55_{\pm0.05}$ | - | - |

Table 2: Performance comparison on Arxiv with full neighborhood.

| METHOD | BATCH SIZE 50 | | | BATCH SIZE 250 | | |
|--------|---------------|--|--|----------------|--|--|
| | AA% ↑ | $\text{AAA}_{val}$% ↑ | AF% ↑ | AA% ↑ | $\text{AAA}_{val}$% ↑ | AF% ↑ |
| BARE | $3.19_{\pm0.50}$ | $11.09_{\pm0.10}$ | $-52.49_{\pm2.06}$ | $4.66_{\pm0.90}$ | $11.45_{\pm0.35}$ | $-41.36_{\pm1.44}$ |
| ER | $5.92_{\pm1.53}$ | $16.96_{\pm0.87}$ | $-49.19_{\pm2.97}$ | $5.30_{\pm0.76}$ | $18.61_{\pm0.67}$ | $-59.95_{\pm1.83}$ |
| EWC | $4.43_{\pm0.33}$ | $10.66_{\pm0.12}$ | $-65.51_{\pm0.22}$ | $2.19_{\pm1.62}$ | $10.96_{\pm1.13}$ | $-26.93_{\pm0.79}$ |
| A-GEM | $16.14_{\pm0.90}$ | $26.10_{\pm0.34}$ | $-41.13_{\pm0.53}$ | $10.61_{\pm1.28}$ | $22.73_{\pm0.43}$ | $-44.83_{\pm1.40}$ |
| LwF | $3.72_{\pm0.71}$ | $11.37_{\pm0.46}$ | $-51.09_{\pm1.60}$ | $4.55_{\pm1.14}$ | $10.88_{\pm0.17}$ | $-50.46_{\pm2.69}$ |
| MAS | $5.25_{\pm0.50}$ | $12.48_{\pm0.80}$ | $-3.07_{\pm0.49}$ | $4.51_{\pm1.14}$ | $13.97_{\pm0.67}$ | $-34.68_{\pm1.03}$ |
| TWP | $2.30_{\pm1.10}$ | $8.77_{\pm1.29}$ | $-21.25_{\pm6.08}$ | $4.43_{\pm0.96}$ | $13.23_{\pm1.88}$ | $-30.94_{\pm3.59}$ |
| JOINT | $46.85_{\pm0.44}$ | - | - | $46.85_{\pm0.44}$ | - | - |

Table 3: Performance comparison on Reddit with full neighborhood.

| METHOD | BATCH SIZE 50 | | | BATCH SIZE 250 | | |
|--------|---------------|--|--|----------------|--|--|
| | AA% ↑ | $\text{AAA}_{val}$% ↑ | AF% ↑ | AA% ↑ | $\text{AAA}_{val}$% ↑ | AF% ↑ |
| BARE | $22.16_{\pm1.26}$ | $39.12_{\pm3.15}$ | $-62.68_{\pm1.59}$ | $21.00_{\pm1.61}$ | $44.66_{\pm3.30}$ | $-73.06_{\pm1.50}$ |
| ER | $33.39_{\pm2.12}$ | $64.11_{\pm0.51}$ | $-64.99_{\pm2.18}$ | $36.93_{\pm1.67}$ | $61.85_{\pm0.46}$ | $-60.66_{\pm1.66}$ |
| EWC | $22.16_{\pm1.26}$ | $39.12_{\pm3.15}$ | $-62.68_{\pm1.59}$ | $18.51_{\pm2.80}$ | $37.65_{\pm4.90}$ | $-67.89_{\pm3.09}$ |
| A-GEM | $57.71_{\pm2.61}$ | $68.22_{\pm0.35}$ | $-36.45_{\pm2.56}$ | $35.54_{\pm1.27}$ | $51.03_{\pm3.54}$ | $-54.60_{\pm1.28}$ |
| LwF | $18.31_{\pm2.34}$ | $41.08_{\pm3.20}$ | $-59.92_{\pm3.50}$ | $21.63_{\pm1.99}$ | $43.73_{\pm2.32}$ | $-68.57_{\pm2.81}$ |
| MAS | $30.06_{\pm1.72}$ | $50.31_{\pm2.18}$ | $-46.72_{\pm1.33}$ | $21.00_{\pm1.61}$ | $44.66_{\pm3.30}$ | $-73.06_{\pm1.50}$ |
| TWP | $22.16_{\pm1.26}$ | $39.12_{\pm3.15}$ | $-62.68_{\pm1.59}$ | $17.64_{\pm2.29}$ | $40.63_{\pm3.17}$ | $-72.79_{\pm2.37}$ |
| JOINT | $90.02_{\pm0.12}$ | - | - | $90.02_{\pm0.12}$ | - | - |

Table 4: Performance comparison on Amazon Computer with full neighborhood.

| METHOD | BATCH SIZE 10 | | | BATCH SIZE 50 | | |
|--------|---------------|--|--|---------------|--|--|
| | AA% ↑ | $\text{AAA}_{val}$% ↑ | AF% ↑ | AA% ↑ | $\text{AAA}_{val}$% ↑ | AF% ↑ |
| BARE | $13.39_{\pm7.46}$ | $40.19_{\pm2.21}$ | $-62.38_{\pm7.69}$ | $18.54_{\pm0.27}$ | $40.76_{\pm0.52}$ | $-74.32_{\pm2.26}$ |
| ER | $27.00_{\pm4.06}$ | $47.48_{\pm0.82}$ | $-65.36_{\pm4.28}$ | $19.74_{\pm0.48}$ | $43.29_{\pm0.59}$ | $-76.45_{\pm0.61}$ |
| EWC | $18.97_{\pm0.16}$ | $41.84_{\pm0.40}$ | $-74.89_{\pm2.68}$ | $18.61_{\pm0.27}$ | $40.78_{\pm0.53}$ | $-74.64_{\pm2.38}$ |
| A-GEM | $35.50_{\pm0.58}$ | $57.22_{\pm0.92}$ | $-57.90_{\pm3.41}$ | $21.38_{\pm0.18}$ | $48.94_{\pm1.03}$ | $-74.58_{\pm1.00}$ |
| LwF | $6.55_{\pm6.47}$ | $36.29_{\pm2.88}$ | $-48.19_{\pm16.77}$ | $3.24_{\pm0.36}$ | $24.60_{\pm1.56}$ | $-21.84_{\pm5.97}$ |
| MAS | $21.85_{\pm4.84}$ | $46.46_{\pm2.32}$ | $-38.97_{\pm9.99}$ | $18.62_{\pm6.09}$ | $44.84_{\pm4.74}$ | $-41.79_{\pm11.41}$ |
| TWP | $18.49_{\pm0.66}$ | $41.27_{\pm0.55}$ | $-72.43_{\pm3.74}$ | $7.78_{\pm6.21}$ | $35.65_{\pm5.47}$ | $-46.67_{\pm16.90}$ |
| JOINT | $72.06_{\pm9.24}$ | - | - | $72.06_{\pm9.24}$ | - | - |

parison of the considered CL methods for the four benchmark datasets is reported in Tables 1-4. All experiments were repeated five times with different initializations, and the metrics are reported as mean and standard deviation across runs. Additionally, we plot in Figure 3 of Appendix D the performance measured using anytime evaluation.

**Final Average Accuracy.** Across all datasets, we observe that none of the compared strategies comes close to the upper bound consisting of joint training. In general, the considered replay methods A-GEM and ER achieve higher final AA compared to the baseline and regularization methods. This can be expected, as rehearsal methods in Continual Learning are generally known to achieve most of the state-of-the-art results (Van De Ven et al., 2022); this holds true also for CGL (Zhang et al., 2022). In particular, A-GEM appears to be the best performing strategy in all cases (except on Reddit when using batch size 250 where it is surpassed by ER), often with a large margin, such as on Arxiv, where it is the only approach with a considerable improvement on the fine-tuned bare model. Regularization methods struggle more, as only on CoraFull EWC and MAS are competitive with A-GEM, while on other datasets their performances are closer to the lower bound.

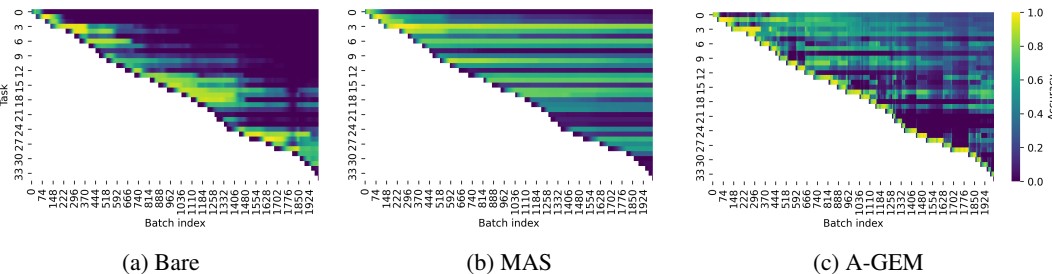

|           |           |           |
|:---------:|:---------:|:---------:|
| (a) Bare  | (b) MAS   | (c) A-GEM |

Figure 1: Anytime evaluation by task: a breakdown of model performance at the end of each training batch for three selected methods on CoraFull with batch size 10.

**Anytime Average Accuracy.** Looking at Average Accuracy gives us an easy way to compare the performance after the entire learning process. However, since in the online setting we expect the model to be ready to make predictions at any time, it is arguably more useful to look at Average Anytime Accuracy as a metric of performance over time. For this purpose Figure 3 (Appendix D) can give us some additional insights into the performance trends through the node stream. We note how it is natural and expected that accuracy tends to decrease with the batch index, as new classes are introduced and the classification task gets increasingly complex. In particular, from Figures 3a and 3b we see that thanks to regularization, on CoraFull the performance of EWC and MAS is much more stable compared to A-GEM, which is much more sensitive to sudden distribution shifts. In this case these two methods might thus be preferable for the stability of their performance. On the other three datasets though the higher stability does not offset their poorer performance. Another phenomenon we note is that while the performance of regularization methods is almost monotonically decreasing, the performance of ER and A-GEM shows much higher variations. We observe abrupt falls at some task boundaries, unexplained by the simple introduction of two new classes and thus due to catastrophic forgetting caused by interference of the current task with old ones. On the other hand, there are also instances where performance increases much more than what could be done with perfect accuracy on the current task, indicating successful backward transfer.

**Forgetting and strategy comparison.** Average Forgetting by itself is not a reliable metric of performance, and must be considered together with accuracy to assess the results of a Continual Learning methods: indeed, not learning anything would lead to 0 forgetting, but this is not our main goal. Indeed, Overall the regularization methods achieve a lower AF score. However, due to how AF is defined, this can be due to lower performances also right after task observation. To better understand the different results in light of the choice of strategy, we report in Figure 1 a more detailed breakdown of accuracy by task for three representative methods. With this comparison we clearly see how the fine-tuned baseline (1a), while it can learn new tasks, is not able to retain past knowledge, which is relatively quickly forgotten. MAS on the other hand (1b), thanks to its regularization is able to preserve quite well the knowledge it has learned, but it struggles to acquire new knowledge as it is presented with more and more classes. Instead A-GEM (1c) seems to achieve a better trade-off between stability and plasticity, maintaining the capacity to learn new tasks while generally preserving acquired knowledge, even showing signs of backward transfer at some task boundaries.

**Impact of batch size.** With regards to the dimensions of the node batches in the stream, we do not observe clear general patterns, as the two considered sizes have somewhat similar results. The only strategy that consistently benefits from a smaller batch size is A-GEM, with considerable increase of performance on all datasets except for CoraFull. For this method (and to a lower extent also for ER), a smaller batch size could thus be a regularizing factor. The fact that very small batch sizes in an online setting can lead to relatively good performance is also encouraging for the future development of OCGL techniques, despite the challenges of catastriphic forgetting (and the online setting for graphs itself, as we discuss in Section 7) that still need to be further addressed.

**Sensitivity to hyperparameters.** In our experiments the backbone model architecture, including number of layers and hidden units, is kept fixed. We conducted an ablation study on CoraFull to assess the impact of these choices, with full results reported in Appendix F. Decreasing the number of hidden units to 128 lowers performances overall, while increasing it to 512 has mixed results, with ER and A-GEM showing small changes, LwF, MAS and TWP scoring lower, but EWC reaches 40% AA. Using 3 GCN layers all results get worse, while with 1 layer there is a general smaller decrease,

except for the bare baseline and ER which improve compared with 2 layers. We note how the difference in results might also be due to a sort of *butterfly effect* caused by the hyperparameter selection policy: since they are selected early on, they could sub-optimal for the entire stream.

# 7 RESULTS ON NEIGHBORHOOD SAMPLING

**Neighborhood expansion.** To assess the neighborhood expansion phenomenon on the four datasets we used in or experiments, we plot in Figure 2 the size of the $l$-hop neighborhood in the evolving graph of each mini-batch in the node stream, for some values of $l$. The Reddit graph in particular is very well connected, with two hops containing the majority of the graph, and three hops practically all nodes. This is the motivation for our use of only one GCN layer on this dataset. CoraFull and Arxiv have a much more contained neighborhood expansion, while Amazon Computer with two hops covers about half of the nodes in the worst cases. Additional plots are shown in Appendix E, with the addition of the number of edges connecting the various hops, which can be used as a proxy of computational complexity.

**Neighborhood sampling.** Following the same experimental setup outlined in Section 5, we conducted experiments with neighborhood sampling instead of using full neighborhood information. As in this case we use sampling to address the problem of neighborhood expansion, we use 2 GCN layers on all datasets. We choose the number of nodes to be sampled with a double rationale: we want to guarantee that processing each mini-batch requires much less than the full graph to conform with the requirements of the online setting, and we want to sample significantly less nodes than the average degree for our analysis of the sampling strategy to be meaningful. Therefore, we fixed the number of neighbors to 5 for CoraFull and Amazon Computer, 10 for Arxiv and 15 for Reddit. Full results are reported in Tables 5-8 and plots with anytime evaluation are shown in Figure 4 (Appendix D). Specifically, on CoraFull only EWC and A-GEM maintain results similar to those obtained with full neighborhood aggregation, while the performance of other models degrades sharply. An Arxiv, where the catastrophic forgetting problem lead to low accuracy scores, the use of sampling does not appear to particularly worsen performance. In fact, the results of ER benefit from it, surpassing A-GEM. The same higher performance of ER is shown on Amazon Computer, where additionally

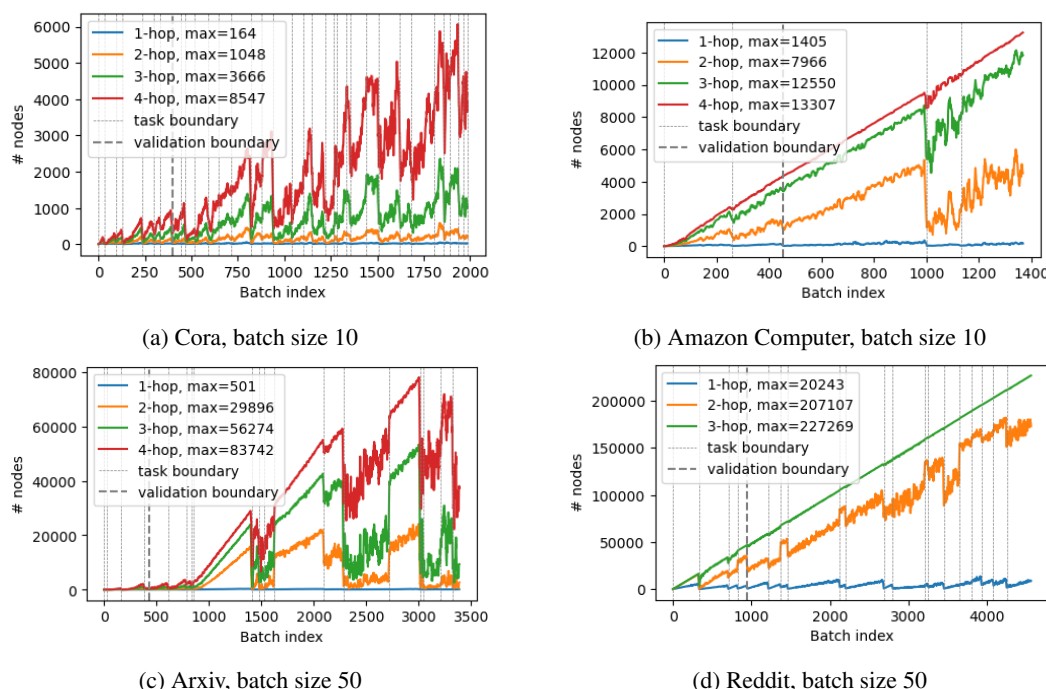

(a) Cora, batch size 10

(b) Amazon Computer, batch size 10

(c) Arxiv, batch size 50

(d) Reddit, batch size 50

Figure 2: Number of nodes in the union of $l$-hop neighborhoods of each training batch. Smoothed with rolling average over windows of 10 batches for readability, maximum is reported in the legend.

Table 5: Performance comparison on CoraFull with neighborhood sampling (5 nodes).

| METHOD | BATCH SIZE 10 | | | BATCH SIZE 50 | | |
|---|---|---|---|---|---|---|
| | AA% ↑ | $AAA_{val}$% ↑ | AF% ↑ | AA% ↑ | $AAA_{val}$% ↑ | AF% ↑ |
| BARE | $8.76_{\pm1.53}$ | $23.47_{\pm2.32}$ | $-30.88_{\pm2.79}$ | $9.80_{\pm1.57}$ | $23.33_{\pm1.32}$ | $-36.59_{\pm2.87}$ |
| ER | $19.17_{\pm1.79}$ | $34.55_{\pm0.69}$ | $-71.14_{\pm2.71}$ | $10.26_{\pm1.79}$ | $25.74_{\pm0.58}$ | $-37.05_{\pm3.22}$ |
| EWC | $26.30_{\pm3.40}$ | $41.46_{\pm1.97}$ | $-21.71_{\pm5.19}$ | $30.30_{\pm4.25}$ | $44.34_{\pm2.89}$ | $-21.07_{\pm4.75}$ |
| A-GEM | $33.08_{\pm0.84}$ | $33.60_{\pm5.88}$ | $-58.78_{\pm0.67}$ | $27.21_{\pm1.26}$ | $37.81_{\pm2.38}$ | $-34.14_{\pm1.97}$ |
| LwF | $14.87_{\pm0.48}$ | $25.92_{\pm0.81}$ | $-49.70_{\pm1.68}$ | $12.01_{\pm0.32}$ | $27.20_{\pm3.49}$ | $-12.97_{\pm1.18}$ |
| MAS | $12.64_{\pm2.32}$ | $39.71_{\pm1.08}$ | $-35.23_{\pm2.27}$ | $26.66_{\pm2.40}$ | $46.70_{\pm0.41}$ | $-32.75_{\pm2.05}$ |
| TwP | $11.73_{\pm1.73}$ | $23.75_{\pm1.75}$ | $-28.58_{\pm4.52}$ | $12.87_{\pm3.14}$ | $25.05_{\pm0.94}$ | $-34.96_{\pm5.11}$ |

Table 6: Performance comparison on Arxiv with neighborhood sampling (10 nodes).

| METHOD | BATCH SIZE 50 | | | BATCH SIZE 250 | | |
|---|---|---|---|---|---|---|
| | AA% ↑ | $AAA_{val}$% ↑ | AF% ↑ | AA% ↑ | $AAA_{val}$% ↑ | AF% ↑ |
| BARE | $4.74_{\pm0.08}$ | $11.88_{\pm0.06}$ | $-82.97_{\pm1.93}$ | $4.67_{\pm0.83}$ | $11.93_{\pm0.08}$ | $-59.34_{\pm2.33}$ |
| ER | $18.41_{\pm2.31}$ | $36.34_{\pm0.26}$ | $-72.96_{\pm2.62}$ | $16.96_{\pm1.45}$ | $32.91_{\pm0.57}$ | $-73.34_{\pm1.60}$ |
| EWC | $4.81_{\pm0.08}$ | $12.35_{\pm0.19}$ | $-77.99_{\pm1.52}$ | $8.49_{\pm3.20}$ | $14.84_{\pm0.38}$ | $-64.19_{\pm4.70}$ |
| A-GEM | $16.43_{\pm3.20}$ | $27.99_{\pm0.52}$ | $-73.36_{\pm3.45}$ | $12.39_{\pm1.14}$ | $21.42_{\pm0.57}$ | $-74.42_{\pm2.18}$ |
| LwF | $4.79_{\pm0.08}$ | $11.85_{\pm0.15}$ | $-79.49_{\pm1.13}$ | $4.48_{\pm1.05}$ | $11.88_{\pm0.15}$ | $-61.81_{\pm1.90}$ |
| MAS | $3.35_{\pm0.99}$ | $12.49_{\pm0.44}$ | $-32.58_{\pm1.73}$ | $4.56_{\pm1.06}$ | $13.48_{\pm0.72}$ | $-49.05_{\pm2.86}$ |
| TWP | $4.74_{\pm0.05}$ | $12.17_{\pm0.14}$ | $-80.22_{\pm0.41}$ | $3.65_{\pm0.94}$ | $11.81_{\pm0.06}$ | $-61.81_{\pm1.90}$ |

Table 7: Performance comparison on Reddit with neighborhood sampling (15 nodes).

| METHOD | BATCH SIZE 50 | | | BATCH SIZE 250 | | |
|---|---|---|---|---|---|---|
| | AA% ↑ | $AAA_{val}$% ↑ | AF% ↑ | AA% ↑ | $AAA_{val}$% ↑ | AF% ↑ |
| BARE | $12.86_{\pm2.56}$ | $36.72_{\pm2.20}$ | $-85.00_{\pm2.54}$ | $20.44_{\pm3.20}$ | $42.20_{\pm3.85}$ | $-58.20_{\pm2.50}$ |
| ER | $17.84_{\pm2.89}$ | $46.24_{\pm0.54}$ | $-81.01_{\pm2.85}$ | $19.18_{\pm3.80}$ | $45.18_{\pm3.94}$ | $-60.14_{\pm5.59}$ |
| EWC | $4.29_{\pm2.73}$ | $20.40_{\pm7.00}$ | $-12.64_{\pm1.54}$ | $5.36_{\pm3.16}$ | $26.56_{\pm6.43}$ | $-14.26_{\pm1.46}$ |
| A-GEM | $43.24_{\pm4.08}$ | $63.44_{\pm3.05}$ | $-55.09_{\pm4.18}$ | $21.97_{\pm4.03}$ | $59.51_{\pm2.36}$ | $-71.29_{\pm3.28}$ |
| LwF | $12.77_{\pm1.69}$ | $37.42_{\pm1.46}$ | $-83.82_{\pm1.89}$ | $16.64_{\pm1.32}$ | $43.40_{\pm1.98}$ | $-76.59_{\pm1.76}$ |
| MAS | $9.91_{\pm0.84}$ | $35.86_{\pm3.21}$ | $-88.15_{\pm0.70}$ | $15.44_{\pm2.54}$ | $37.62_{\pm3.97}$ | $-80.23_{\pm2.94}$ |
| TWP | $12.60_{\pm2.13}$ | $36.10_{\pm0.26}$ | $-85.46_{\pm2.23}$ | $20.16_{\pm4.52}$ | $40.96_{\pm3.84}$ | $-59.25_{\pm7.10}$ |

Table 8: Performance comparison on Amazon Computer with neighborhood sampling (5 nodes).

| METHOD | BATCH SIZE 10 | | | BATCH SIZE 50 | | |
|---|---|---|---|---|---|---|
| | AA% ↑ | $AAA_{val}$% ↑ | AF% ↑ | AA% ↑ | $AAA_{val}$% ↑ | AF% ↑ |
| BARE | $19.34_{\pm0.26}$ | $43.03_{\pm0.15}$ | $-78.66_{\pm0.35}$ | $18.47_{\pm0.76}$ | $41.65_{\pm0.33}$ | $-77.04_{\pm1.71}$ |
| ER | $24.37_{\pm2.03}$ | $52.34_{\pm2.30}$ | $-59.93_{\pm7.94}$ | $48.28_{\pm7.83}$ | $67.00_{\pm0.71}$ | $-45.77_{\pm9.46}$ |
| EWC | $17.60_{\pm2.53}$ | $40.12_{\pm1.60}$ | $-35.37_{\pm9.96}$ | $19.39_{\pm0.18}$ | $43.32_{\pm0.19}$ | $-78.15_{\pm1.30}$ |
| A-GEM | $20.05_{\pm0.59}$ | $50.36_{\pm1.35}$ | $-77.13_{\pm0.96}$ | $19.95_{\pm1.16}$ | $50.28_{\pm1.24}$ | $-77.50_{\pm0.52}$ |
| LwF | $19.29_{\pm0.48}$ | $42.95_{\pm0.19}$ | $-78.35_{\pm0.50}$ | $18.12_{\pm0.62}$ | $41.09_{\pm0.67}$ | $-76.37_{\pm2.36}$ |
| MAS | $18.82_{\pm1.19}$ | $42.92_{\pm1.83}$ | $-60.99_{\pm7.52}$ | $18.66_{\pm0.18}$ | $43.17_{\pm0.61}$ | $-77.33_{\pm0.72}$ |
| TWP | $19.13_{\pm0.45}$ | $42.98_{\pm0.18}$ | $-78.54_{\pm0.39}$ | $17.82_{\pm0.45}$ | $41.78_{\pm1.01}$ | $-71.46_{\pm3.48}$ |

we do not see the same low performance of LwF as without sampling. Finally, on Reddit the performance degradation is more pronounced, as the number of neighbors was cut more substantially. In summary, as expected due to ignoring some neighborhood information, most of the performances are lowered, indicating that more research is required to properly address this issue in OCGL.

## 8 CONCLUSIONS

In this paper, we introduced the formulation of the Online Continual Graph Learning setting, closing the gap between the Continual Graph Learning and Online Continual Learning literature. We adapted four node classification datasets to the proposed framework, constructing node streams starting from the class-incremental learning scenario. Our evaluation of suitably adapted Continual Learning methods highlights the higher performance of replay-based methods. Finally, we raise the issue of neighborhood expansion for GNNs, proposing neighborhood sampling as a straight-forward solution to bound the computational cost of training on each mini-batch. In future works, we plan to further tackle the issue of neighborhood expansion, developing tailored strategies that can ensure computational efficiency while better addressing the catastrophic forgetting problem. We further intend to consider more diverse node stream construction and additional tasks such as link prediction.

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

## A    DATASETS

In the experiments for this paper, we used four node-level classification graph datasets. The CoraFull dataset (Bojchevski & Günnemann, 2018) is a citation network where nodes represent research papers and edges indicate citations between them, with labels based on paper topics. Arxiv (Hu et al., 2021) is a larger citation network derived from arXiv papers in the Computer Science category. The Reddit dataset (Hamilton et al., 2017) consists of posts from different communities of the Reddit platform, where nodes represent posts, and edges connect posts commented on by the same user, forming a large interaction graph. Finally, Amazon Computer (Shchur et al., 2019) is a co-purchase network, where nodes are products and edges indicate frequently co-purchased items within the computers category on Amazon. Summary statistics for the four graphs are reported in Table 9.

Table 9: Dataset statistics.

| DATASET | CORAFULL | ARXIV | REDDIT | AMAZON COMPUTER |
|---|---|---|---|---|
| # NODES | 19,793 | 169,343 | 227,853 | 13,752 |
| # EDGES | 130,622 | 1,166,243 | 114,615,892 | 491,722 |
| # CLASSES | 70 | 40 | 40 | 10 |

## B    METRICS

Thanks to the construction of the node stream starting from the class-incremental setting, we can use two widely used metrics in CL: *Average Accuracy (AA)* and *Average Forgetting (AF)* (Lopez-Paz & Ranzato, 2017). The most comprehensive metric for CL, from which AA and AF are derived, is the performance matrix $M \in \mathbb{R}^{T \times T}$, where $T$ is the number of tasks and $M_{i,j}$ is the test classification accuracy on task $j$ after the model has observed task $i$. AA is then defined as $\text{AA} = \frac{1}{T} \sum_{i=1}^{T} M_{T,i}$, and average forgetting as $\text{AF} = \frac{1}{T-1} \sum_{i=1}^{T-1} M_{T,i} - M_{i,i}$. AA serves as a single value to quantify the performance of the model after having observed the entire sequence of tasks, or stream in our case. AF measures the performance degradation (forgetting), that occurs from when a task was just observed to the end of training.

To assess the performance of the model throughout the node stream, we also perform anytime evaluation, meaning that we evaluate the model on validation nodes after training on each mini-batch (Koh et al., 2021). This allows us to capture the performance at any point in time, and observe also graphically how the model reacts to changes in data distribution. We measure this with *Average Anytime Accuracy (AAA)* (Caccia et al., 2021), which is a generalization of average incremental accuracy for the online setting. Indicating with $\text{AA}_t$ the average accuracy after training on batch $t$, and having $n$ batches in total, AAA is defined as $\text{AAA} = \frac{1}{n} \sum_{t=1}^{n} \text{AA}_t$. This can be interpreted as an Area Under the Curve accuracy score (Koh et al., 2021).

## C  HYPERPARAMETERS

A standard grid search was performed to select training hyperparameters for the models used in all experiments. We detail here the specific search space for each of the methods used in our comparisons. Two hyperparameters are common for all techniques: the learning rate, selected in the set $\{0.01, 0.001, 0.0001, 0.00001\}$, and the number of passes on each batch before passing to the next one, chosen between 1 and 5. No weight decay or dropout were used. Method specific hyperparameters are reported in Table 10, and specific details can be found in the original papers. In particular, the hyperparameters of regularization methods regulate the strenght of the regularization. For LwF a new hyperparameter has been introduced to adapt to the online setting: the number of batches after which to update the teacher model. For replay based methods we consider memory size (budget) and the proportion of memories to use with respect to each training batch.

Table 10: Method specific hyperparameters.

| METHOD | HYPERPARAMETER CANDIDATES |
|---|---|
| ER | BUDGET: $\{100, 1000\}$; MEMORY_PROPORTION: $\{1,2,3\}$ |
| EWC | LAMBDA: $\{10^0, 10^2, 10^4, 10^6, 10^8, 10^{10}\}$ |
| A-GEM | BUDGET: $\{100, 1000\}$; MEMORY_PROPORTION: $\{1,2,3\}$ |
| LwF | LAMBDA_DIST: $\{0.1,1,10\}$; T: $\{0.2,2,20\}$, UPDATE_EVERY: $\{1, 10, 100\}$ |
| MAS | LAMBDA: $\{10^0, 10^2, 10^4, 10^6, 10^8, 10^{10}\}$ |
| TwP | LAMBDA_L: $\{10^2, 10^4, 10^6\}$; LAMBDA_T: $\{10^2, 10^4, 10^6\}$; BETA: $\{0.001, 0.01, 0.1\}$ |

## D  ANYTIME EVALUATION PLOTS

We show here the plots with anytime evaluation, on all four datasets and with both choices of batch size. In Figure 3 we show the results using full neighborhood information (see Section 6 for all the results and comments), and in Figure 4 the anytime evaluation when using neighborhood sampling (more details in Section 7 of the main paper).

## E  NEIGHBORHOOD EXPANSION

Neighborhood expansion has been identified as the main issue with the online setting fro graphs. In Figure 2 we showed the phenomenon, and we report here additional plots for all datasets and batch sizes. In particular, in Figures 5-7 we provide measure neighborhood expansion non only in term of number of nodes in the $l$-hop neighborhood, as done in Section 7, but we additionally count the number of edges between hops, which gives us a good proxy of computational time. Interestingly, the number of edges increases more drastically than the number of nodes when incrementing the number of layers, further confirming the need for a strategy such as sampling to maintain the computational complexity bounded within the limits of the online learning setting.

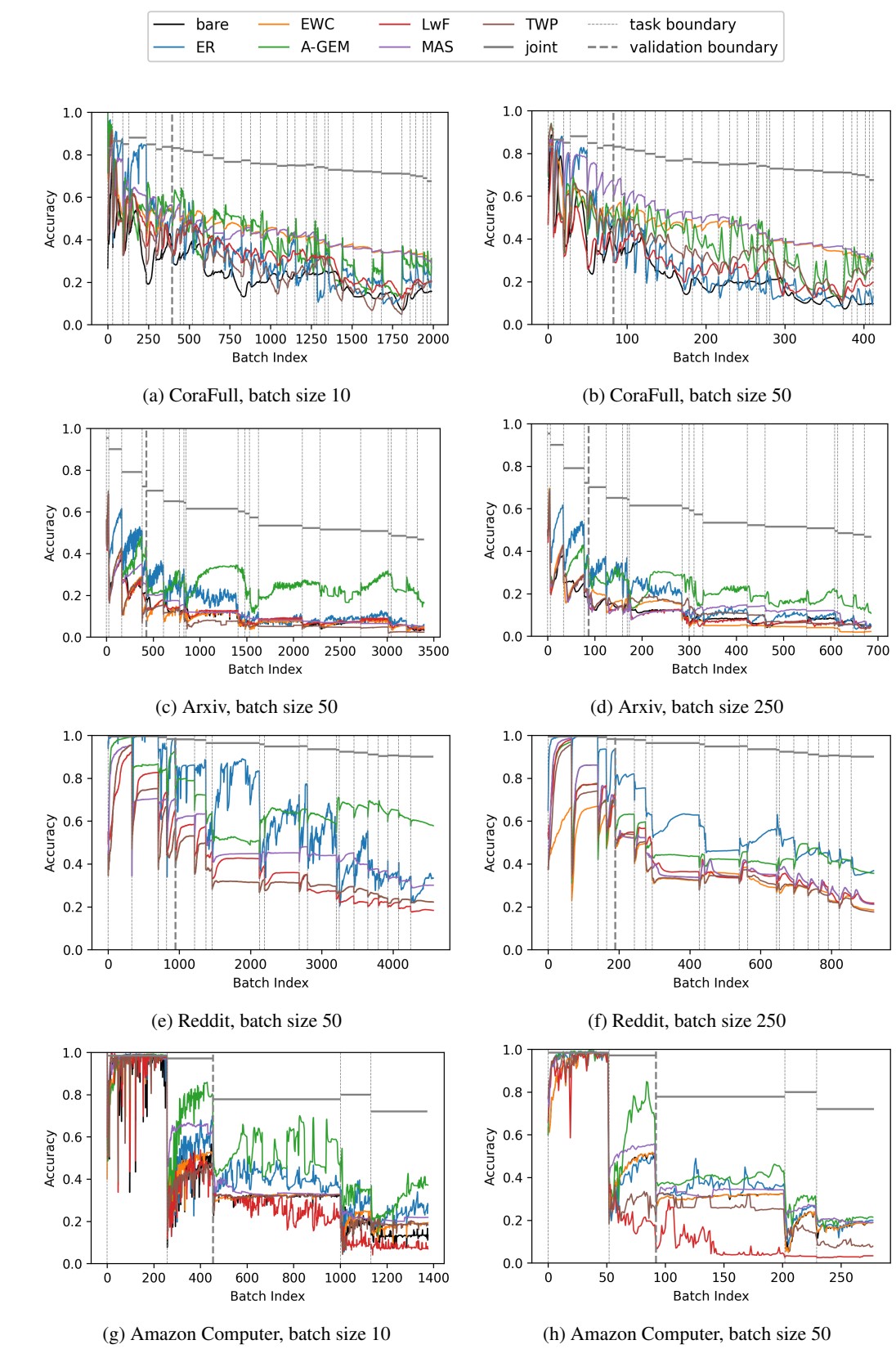

Figure 3: Anytime evaluation, showing AA on validation nodes after training on each mini-batch. We highlight the boundaries between tasks and the threshold up to which hyperparameter selection is performed.

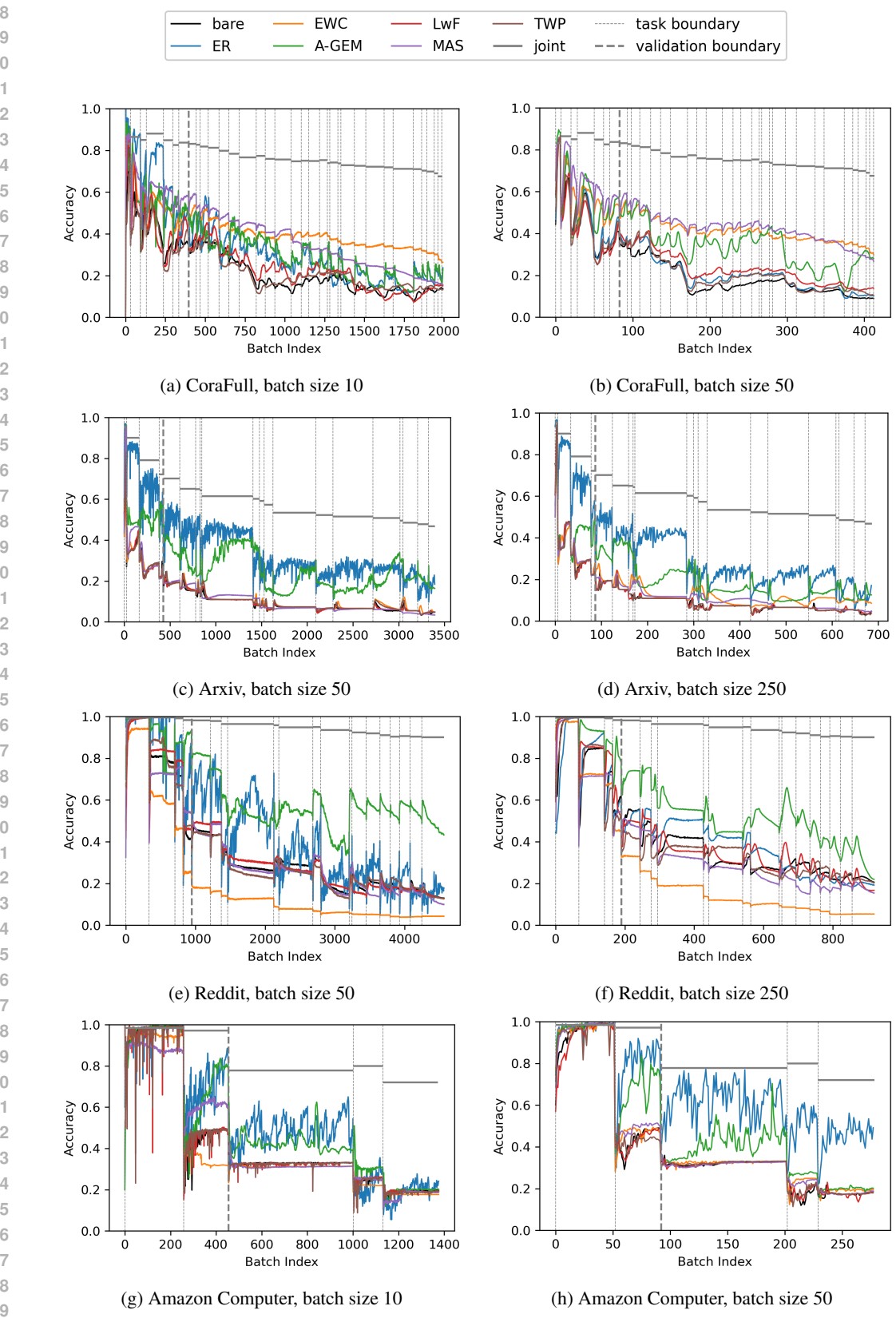

Figure 4: Anytime evaluation, showing AA on validation nodes after training on each mini-batch, when performing neighborhood sampling. We highlight the boundaries between tasks and the threshold up to which hyperparameter selection is performed.

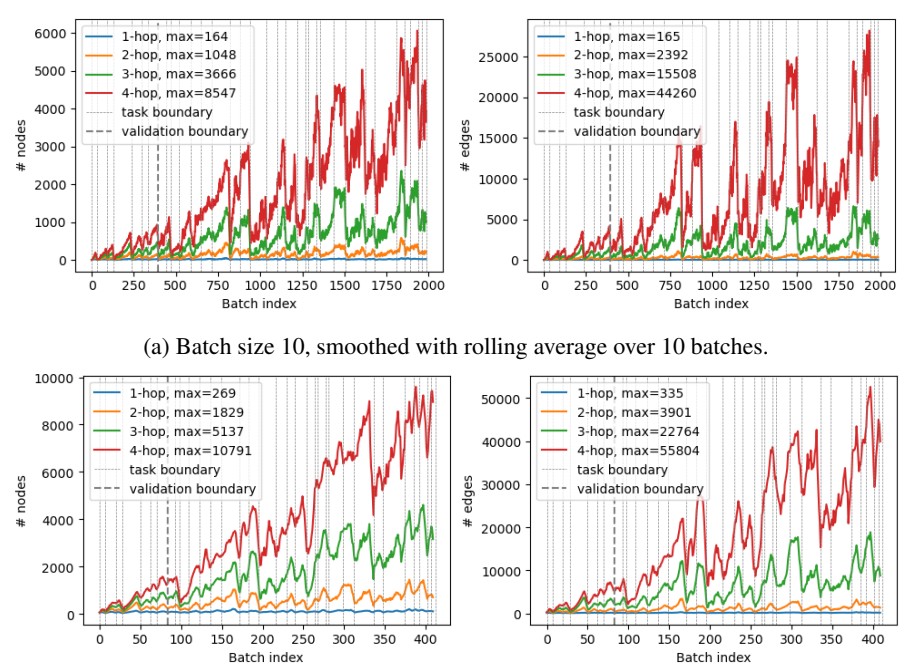

(a) Batch size 10, smoothed with rolling average over 10 batches.

(b) Batch size 50, smoothed with rolling average over 5 batches.

Figure 5: Neighborhood Expansion on CoraFull dataset (total nodes: 19,793). On the left: number of nodes in the l-hop neighborhood of the training batch; on the right: number of edges connecting the l-1 to l-hop neighborhood of the training batch.

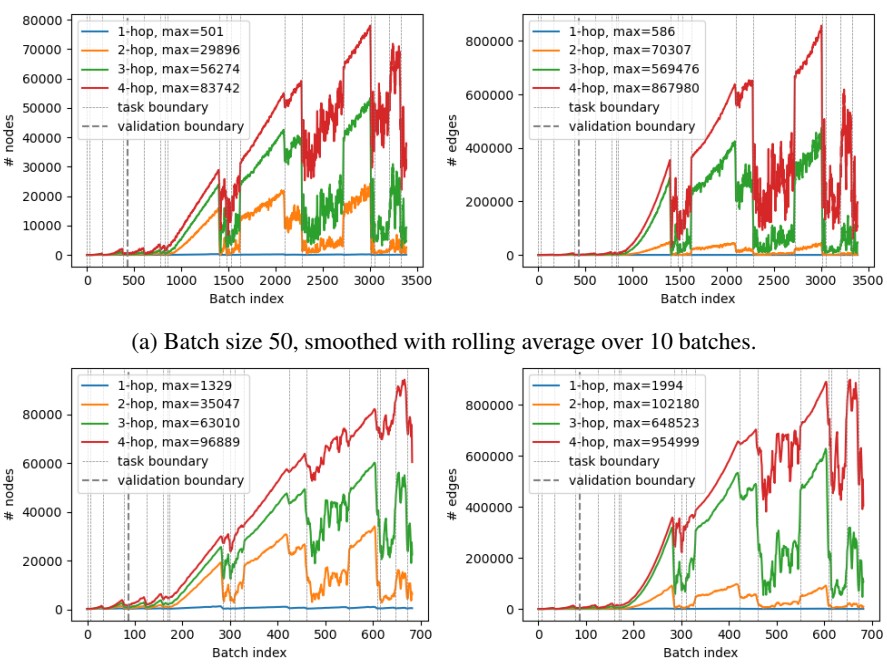

(a) Batch size 50, smoothed with rolling average over 10 batches.

(b) Batch size 250, smoothed with rolling average over 5 batches.

Figure 6: Neighborhood Expansion on Arxiv dataset (total nodes: 169,343). On the left: number of nodes in the l-hop neighborhood of the training batch; on the right: number of edges connecting the l-1 to l-hop neighborhood of the training batch.

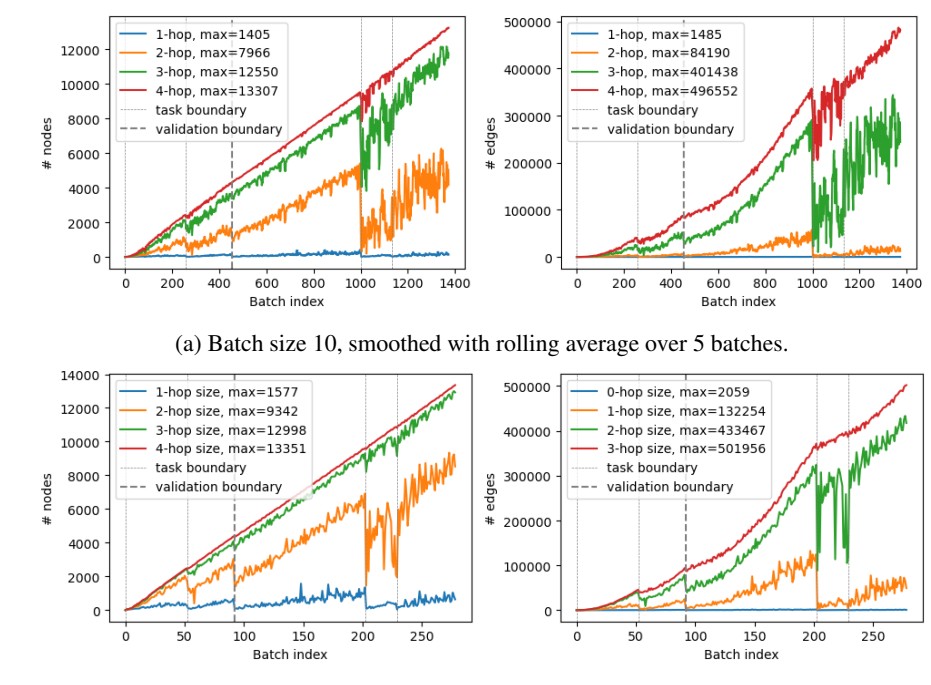

(a) Batch size 10, smoothed with rolling average over 5 batches.

(b) Batch size 50 (no smoothing).

Figure 7: Neighborhood Expansion on Amazon Computer dataset (total nodes: 13,752). On the left: number of nodes in the l-hop neighborhood of the training batch; on the right: number of edges connecting the l-1 to l-hop neighborhood of the training batch.

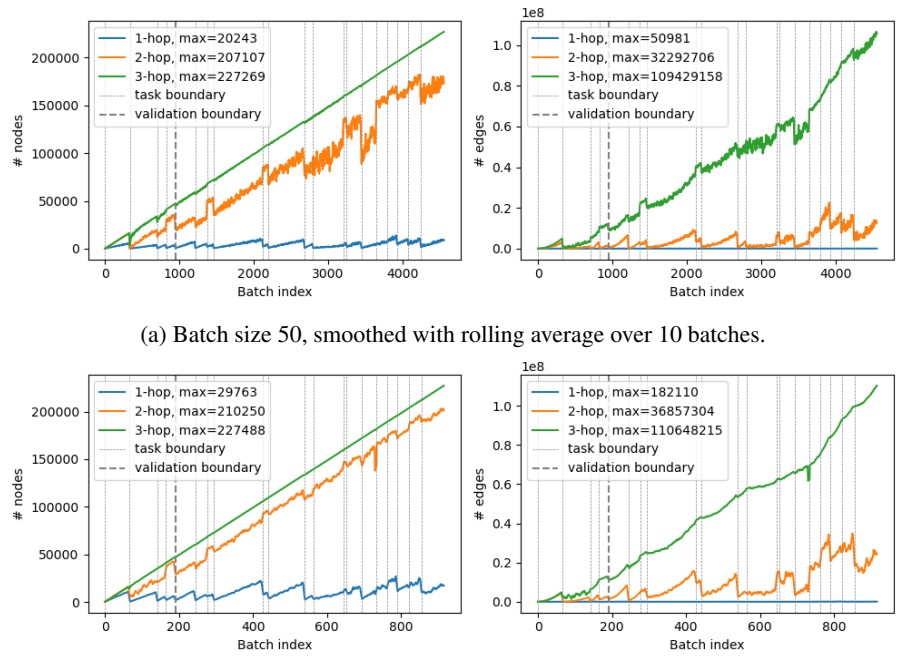

(a) Batch size 50, smoothed with rolling average over 10 batches.

(b) Batch size 250, smoothed with rolling average over 5 batches.

Figure 8: Neighborhood Expansion on Reddit dataset (total nodes: 227,853). On the left: number of nodes in the l-hop neighborhood of the training batch; on the right: number of edges connecting the l-1 to l-hop neighborhood of the training batch.

# F ABLATION STUDY ON CORAFULL

In our experiments, the backbone model for all Continual Learning strategies was fixed as a 2-layer GCN with 256 hidden units (with the exception of the Reddit dataset where with full neighborhood only one layer was used), following Zhang et al. (2022). Thus, we performed a small ablation study on the CoraFull dataset to assess the impact of the number of hidden units and number of GCN layers. This is conducted only with full neighborhood aggregation.

## F.1 NUMBER OF HIDDEN UNITS

Still maintaining a 2-layer GCN, we changed the number of hidden units, from the original 256 to alternatively 128 and 512. We report the results in Tables 11-12, and we show anytime evaluation in Figure 9. We observe how overall performance are lower than with 256 hidden units, possibly as this number may have been validated in previous works. A smaller network seems to consistently damage performance metrics, while with 512 units A-GEM scores similarly as before, and EWC actually improves greatly, reaching 40% AA. If the increase in computational complexity with additional units is not significant, it may thus be worthwhile to consider validating the number of hidden units as well.

Table 11: Performance comparison on CoraFull with 512 hidden units.

| METHOD | BATCH SIZE 10 | | | BATCH SIZE 50 | | |
|---|---|---|---|---|---|---|
| | AA% ↑ | AAA$_{val}$% ↑ | AF% ↑ | AA% ↑ | AAA$_{val}$% ↑ | AF% ↑ |
| BARE | $18.16_{\pm0.57}$ | $26.30_{\pm0.53}$ | $-45.41_{\pm0.98}$ | $9.94_{\pm0.40}$ | $25.30_{\pm0.30}$ | $-35.66_{\pm1.58}$ |
| ER | $27.09_{\pm2.16}$ | $35.20_{\pm0.29}$ | $-64.72_{\pm2.34}$ | $10.28_{\pm0.51}$ | $25.49_{\pm0.41}$ | $-36.63_{\pm1.47}$ |
| EWC | $38.56_{\pm1.80}$ | $48.70_{\pm1.01}$ | $-22.13_{\pm2.76}$ | $40.76_{\pm1.05}$ | $52.15_{\pm0.80}$ | $-19.52_{\pm2.10}$ |
| A-GEM | $34.57_{\pm2.03}$ | $37.73_{\pm0.35}$ | $-56.61_{\pm2.19}$ | $28.59_{\pm1.92}$ | $42.51_{\pm0.38}$ | $-16.95_{\pm1.35}$ |
| LwF | $19.61_{\pm0.84}$ | $26.41_{\pm0.71}$ | $-43.19_{\pm1.26}$ | $12.93_{\pm1.42}$ | $27.73_{\pm0.26}$ | $-36.52_{\pm1.35}$ |
| MAS | $3.10_{\pm0.31}$ | $26.61_{\pm1.38}$ | $-27.12_{\pm1.50}$ | $11.52_{\pm1.62}$ | $37.41_{\pm1.42}$ | $-23.65_{\pm1.91}$ |
| TWP | $18.18_{\pm0.57}$ | $26.30_{\pm0.53}$ | $-45.40_{\pm0.99}$ | $9.90_{\pm0.41}$ | $25.29_{\pm0.29}$ | $-35.69_{\pm1.48}$ |

Table 12: Performance comparison on CoraFull with 128 hidden units.

| METHOD | BATCH SIZE 10 | | | BATCH SIZE 50 | | |
|---|---|---|---|---|---|---|
| | AA% ↑ | AAA$_{val}$% ↑ | AF% ↑ | AA% ↑ | AAA$_{val}$% ↑ | AF% ↑ |
| BARE | $15.12_{\pm2.27}$ | $22.75_{\pm1.64}$ | $-35.08_{\pm2.38}$ | $10.74_{\pm0.97}$ | $24.55_{\pm1.10}$ | $-25.33_{\pm1.83}$ |
| ER | $15.05_{\pm2.38}$ | $29.55_{\pm0.40}$ | $-66.55_{\pm1.51}$ | $13.96_{\pm1.18}$ | $32.48_{\pm0.33}$ | $-70.49_{\pm1.33}$ |
| EWC | $5.73_{\pm1.47}$ | $21.50_{\pm3.44}$ | $-6.50_{\pm2.53}$ | $5.90_{\pm1.73}$ | $28.90_{\pm2.14}$ | $-20.11_{\pm4.39}$ |
| A-GEM | $15.94_{\pm3.23}$ | $28.41_{\pm0.65}$ | $-71.24_{\pm2.38}$ | $26.62_{\pm1.48}$ | $39.37_{\pm0.47}$ | $-34.64_{\pm1.77}$ |
| LWF | $15.30_{\pm1.61}$ | $26.03_{\pm0.28}$ | $-69.40_{\pm0.69}$ | $10.06_{\pm2.76}$ | $24.78_{\pm0.85}$ | $-29.47_{\pm2.63}$ |
| MAS | $6.25_{\pm2.74}$ | $28.81_{\pm1.89}$ | $-24.70_{\pm2.94}$ | $10.85_{\pm3.74}$ | $33.50_{\pm1.97}$ | $-31.71_{\pm3.37}$ |
| TwP | $15.12_{\pm2.27}$ | $22.75_{\pm1.64}$ | $-35.10_{\pm2.35}$ | $10.70_{\pm0.96}$ | $24.56_{\pm1.10}$ | $-25.33_{\pm1.81}$ |

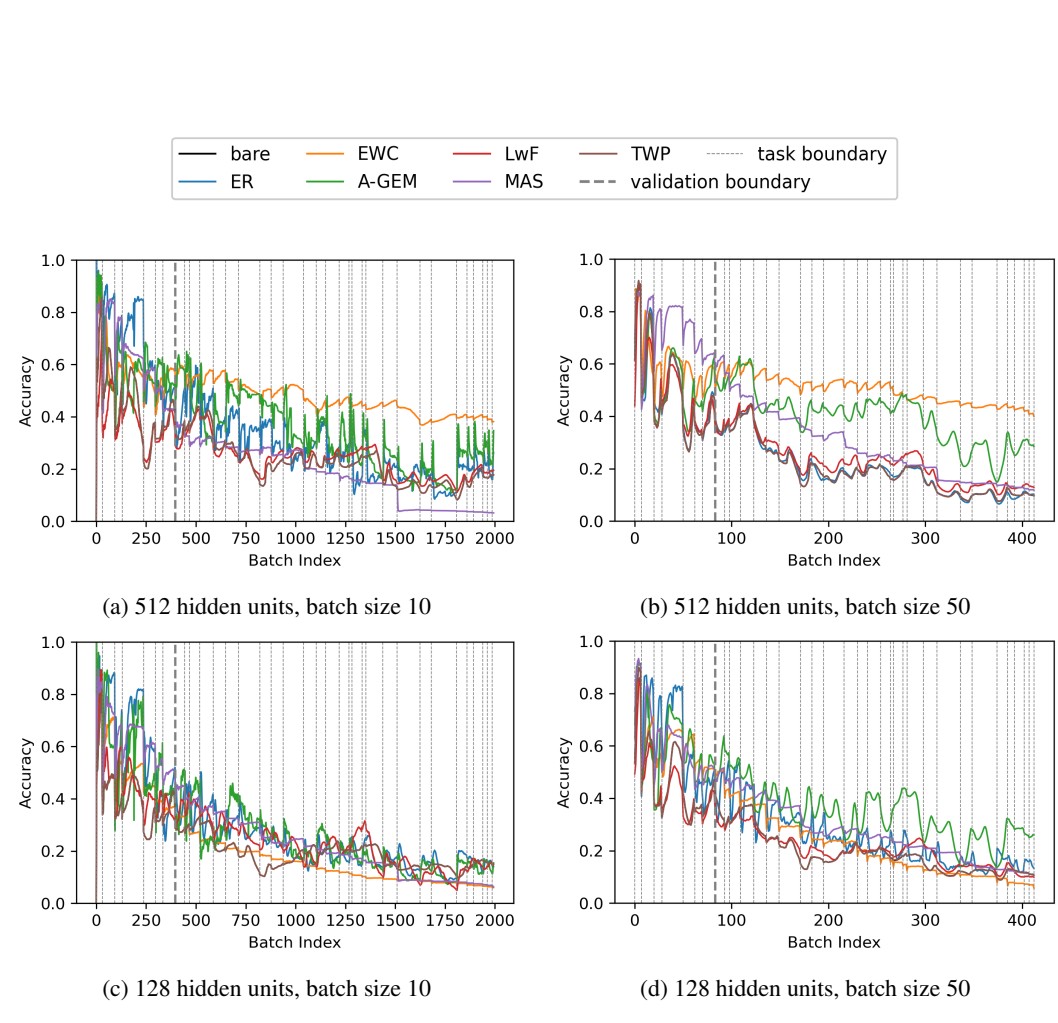

(a) 512 hidden units, batch size 10

(b) 512 hidden units, batch size 50

(c) 128 hidden units, batch size 10

(d) 128 hidden units, batch size 50

Figure 9: Anytime evaluation on CoraFull, showing AA on validation nodes after training on each mini-batch. We highlight the boundaries between tasks and the threshold up to which hyperparameter selection is performed.

## F.2 NUMBER OF GCN LAYERS

We considered here a different number of GCN layers compared to our main results, 1 and 3 specifically, as on CoraFull also the 3-hop neighborhood has a relatively limited expansion. Results are reported in Tables 13-14, and anytime evaluation plots in Figure 10. We observe overall lower results specifically with 3 layers, while ER improves using only 1 GCN layer.

Additionally, as a sort of "0-layer" baseline, we test the usage of a 2 layer MLP, equivalent to our main backbone GCN applied using the identity as adjacency matrix. This is the same as discarding graph topology information. While most results are consequently greatly reduced, ER, GEM and EWC are able to maintain some performance even without using the graph topology.

Table 13: Performance comparison on CoraFull with 1 GCN layer.

| METHOD | BATCH SIZE 10 | | | BATCH SIZE 50 | | |
| --- | --- | --- | --- | --- | --- | --- |
| | AA% ↑ | $AAA_{val}$% ↑ | AF% ↑ | AA% ↑ | $AAA_{val}$% ↑ | AF% ↑ |
| BARE | $18.39_{\pm 0.06}$ | $24.02_{\pm 0.02}$ | $-76.21_{\pm 0.08}$ | $15.00_{\pm 0.10}$ | $22.94_{\pm 0.15}$ | $-27.09_{\pm 0.19}$ |
| ER | $38.68_{\pm 0.67}$ | $57.97_{\pm 0.40}$ | $-53.69_{\pm 0.73}$ | $33.38_{\pm 0.83}$ | $55.66_{\pm 0.18}$ | $-60.26_{\pm 0.72}$ |
| EWC | $18.40_{\pm 0.11}$ | $24.01_{\pm 0.03}$ | $-76.14_{\pm 0.08}$ | $15.01_{\pm 0.15}$ | $22.87_{\pm 0.11}$ | $-27.30_{\pm 0.22}$ |
| A-GEM | $28.90_{\pm 1.15}$ | $54.16_{\pm 0.26}$ | $-64.51_{\pm 1.42}$ | $29.21_{\pm 0.86}$ | $52.70_{\pm 0.11}$ | $-64.92_{\pm 1.00}$ |
| LwF | $24.09_{\pm 0.34}$ | $28.77_{\pm 0.10}$ | $-64.59_{\pm 0.41}$ | $15.08_{\pm 0.13}$ | $22.88_{\pm 0.19}$ | $-27.26_{\pm 0.19}$ |
| MAS | $34.49_{\pm 0.08}$ | $31.93_{\pm 0.03}$ | $-58.54_{\pm 0.08}$ | $15.01_{\pm 0.15}$ | $22.87_{\pm 0.11}$ | $-27.30_{\pm 0.22}$ |
| TWP | $17.01_{\pm 0.14}$ | $23.36_{\pm 0.01}$ | $-76.52_{\pm 0.11}$ | $15.01_{\pm 0.05}$ | $22.85_{\pm 0.13}$ | $-26.97_{\pm 0.34}$ |

Table 14: Performance comparison on CoraFull with 3 GCN layers.

| METHOD | BATCH SIZE 10 | | | BATCH SIZE 50 | | |
| --- | --- | --- | --- | --- | --- | --- |
| | AA% ↑ | $AAA_{val}$% ↑ | AF% ↑ | AA% ↑ | $AAA_{val}$% ↑ | AF% ↑ |
| BARE | $6.66_{\pm 1.83}$ | $18.24_{\pm 0.13}$ | $-83.82_{\pm 1.84}$ | $5.99_{\pm 0.49}$ | $17.70_{\pm 0.82}$ | $-52.80_{\pm 1.36}$ |
| ER | $9.32_{\pm 1.41}$ | $24.08_{\pm 0.56}$ | $-77.20_{\pm 1.76}$ | $2.74_{\pm 0.21}$ | $16.40_{\pm 1.04}$ | $-32.44_{\pm 14.08}$ |
| EWC | $18.20_{\pm 2.25}$ | $38.86_{\pm 2.95}$ | $-25.36_{\pm 2.71}$ | $20.89_{\pm 3.20}$ | $43.46_{\pm 2.09}$ | $-23.17_{\pm 4.05}$ |
| A-GEM | $3.85_{\pm 0.13}$ | $19.97_{\pm 0.62}$ | $-61.79_{\pm 11.32}$ | $23.91_{\pm 3.51}$ | $34.53_{\pm 0.90}$ | $-29.78_{\pm 3.63}$ |
| LwF | $16.18_{\pm 1.45}$ | $30.47_{\pm 1.40}$ | $-53.27_{\pm 2.98}$ | $15.69_{\pm 1.90}$ | $28.96_{\pm 1.26}$ | $-42.63_{\pm 4.37}$ |
| MAS | $13.57_{\pm 1.83}$ | $33.62_{\pm 2.02}$ | $-21.95_{\pm 2.53}$ | $28.36_{\pm 1.24}$ | $49.34_{\pm 1.51}$ | $-11.68_{\pm 1.47}$ |
| TWP | $12.62_{\pm 2.58}$ | $21.91_{\pm 0.73}$ | $-58.41_{\pm 3.34}$ | $7.86_{\pm 3.14}$ | $18.87_{\pm 1.25}$ | $-17.47_{\pm 4.27}$ |

Table 15: Performance comparison on CoraFull with MLP (no graph structure).

| METHOD | BATCH SIZE 10 | | | BATCH SIZE 50 | | |
| --- | --- | --- | --- | --- | --- | --- |
| | AA% ↑ | $AAA_{val}$% ↑ | AF% ↑ | AA% ↑ | $AAA_{val}$% ↑ | AF% ↑ |
| BARE | $2.90_{\pm 1.07}$ | $16.14_{\pm 1.17}$ | $-36.09_{\pm 2.86}$ | $2.62_{\pm 0.58}$ | $13.22_{\pm 0.24}$ | $-20.90_{\pm 0.53}$ |
| ER | $25.25_{\pm 0.63}$ | $33.18_{\pm 0.34}$ | $-67.41_{\pm 0.51}$ | $13.61_{\pm 1.15}$ | $31.74_{\pm 0.12}$ | $-76.10_{\pm 1.10}$ |
| EWC | $18.28_{\pm 3.78}$ | $30.83_{\pm 2.21}$ | $-38.09_{\pm 4.47}$ | $2.22_{\pm 0.91}$ | $18.50_{\pm 0.63}$ | $-45.75_{\pm 2.93}$ |
| GEM | $26.56_{\pm 1.70}$ | $38.95_{\pm 0.11}$ | $-45.83_{\pm 1.85}$ | $3.10_{\pm 0.22}$ | $16.80_{\pm 0.17}$ | $-20.85_{\pm 0.49}$ |
| LWF | $5.65_{\pm 1.09}$ | $18.34_{\pm 0.68}$ | $-35.84_{\pm 2.24}$ | $3.53_{\pm 0.48}$ | $14.69_{\pm 0.33}$ | $-22.45_{\pm 1.18}$ |
| MAS | $2.69_{\pm 0.20}$ | $20.41_{\pm 0.36}$ | $-46.12_{\pm 0.73}$ | $10.78_{\pm 2.29}$ | $26.87_{\pm 0.58}$ | $-42.59_{\pm 2.54}$ |
| TWP | $4.98_{\pm 1.28}$ | $17.42_{\pm 0.92}$ | $-37.57_{\pm 4.05}$ | $2.57_{\pm 0.25}$ | $13.03_{\pm 0.39}$ | $-21.42_{\pm 0.97}$ |

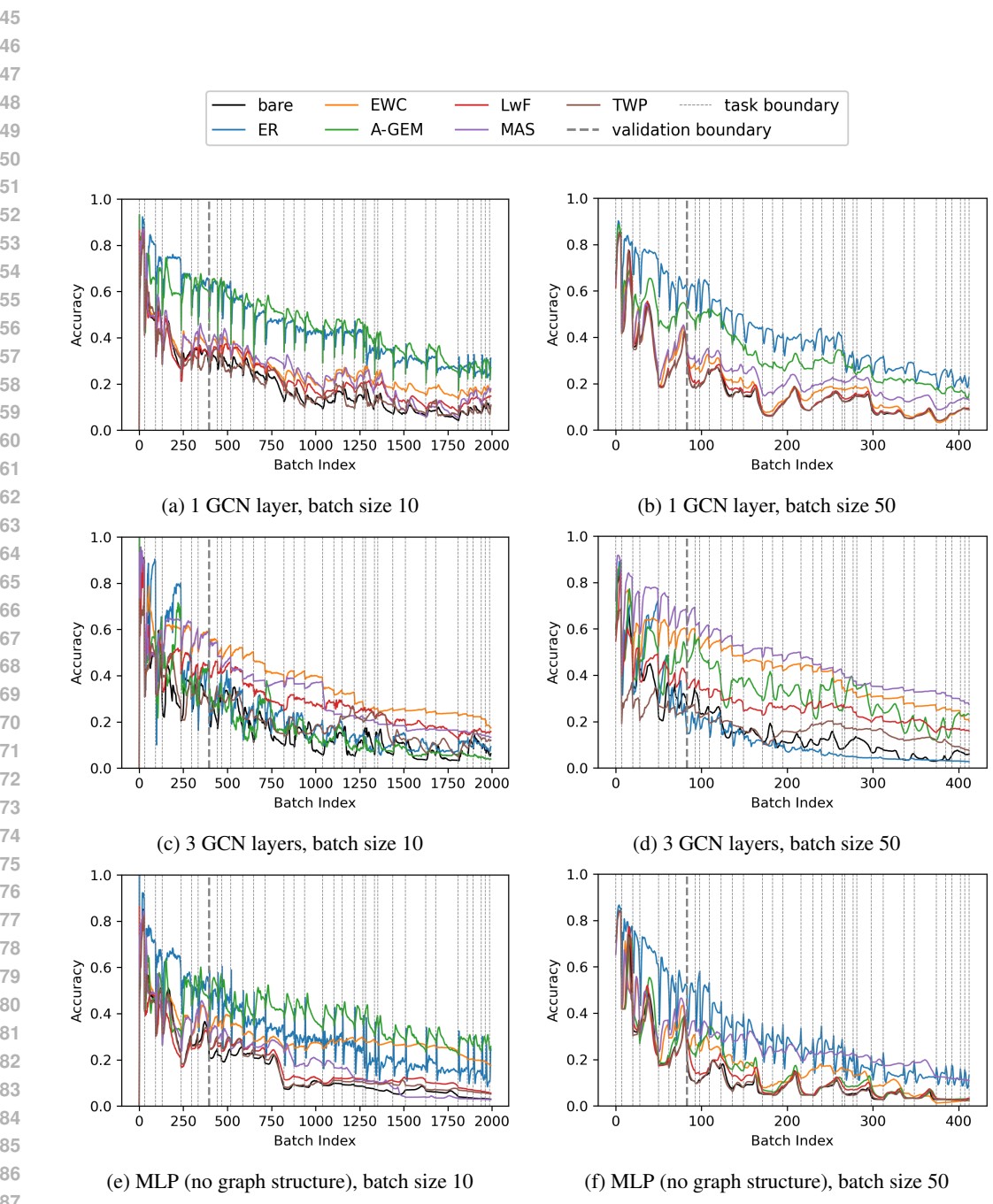

Figure 10: Anytime evaluation on CoraFull, showing AA on validation nodes after training on each mini-batch. We highlight the boundaries between tasks and the threshold up to which hyperparameter selection is performed.

