# OpenReview forum: "Online Continual Graph Learning"
_ICLR.cc/2025/Conference — Submitted to ICLR 2025_

### Official Review · Reviewer_n8Q1 · 2024-10-26

**Soundness:** 2
**Presentation:** 2
**Contribution:** 1
**Rating:** 3
**Confidence:** 4

**Summary:**

The paper introduces an Online Continual Graph Learning (OCGL) framework designed for learning in dynamic graph environments where data arrives in a streaming fashion. The authors address challenges of catastrophic forgetting in graph-based continual learning, particularly when working with graph neural networks (GNNs) that rely on neighborhood information, which can lead to high memory and computational costs. The proposed OCGL framework is evaluated on four node classification datasets, using modified continual learning methods suited for online learning. They propose neighborhood sampling as a strategy to address neighborhood expansion challenges, which could otherwise lead to prohibitive costs in dynamic graph settings.

**Strengths:**

- the studied problem is an important research question

- I like the attempts that the author tried to take a more systematic approach toward the problem

**Weaknesses:**

1. the technical content of the paper does not address the research question proposed by the authors. For example
    a. one of the claimed distributions is the formulation of the so-called online GCL. However, I do not see any formal formulation of the problem. Only a generic description is provided. For example, to fulfil the claim, one would naturally expect to see the data model, the definition and format of the leaner, and the properties and requirements for an effective learner in this scenario. None of these information is provided.
   b. another claim is that online GCL is a new learning paradigm and different from GCL. One would expect to see a detailed comparison between these two. How are they different exactly? How much is the difference?

2. there are some statements that are not factually correct. For example, continual learning is inherently an online setting. An ideal continual learning algorithm should adaptively learn from the new data without the need to access previous data. However, this can be proved theoretically impossible. Therefore, many continual learning algorithms compromise by allowing partial access to historical data. Even for online systems, storing historical data is also allowed. Regarding the task boundary, there have been many studies that looked at the continual learning setting without a clear task boundary. These studies have been under the terms such as "task-free continual learning" and "domain-free continual learning"[1]. Furthermore, it is not clear what exactly task-boundary means in the paper.

3. the proposed technique is standard and the paper has no conceivable novelty or contribution. The neighbourhood explosion problem is a standard issue in GNN training even for the case of training GNN on static graphs and neighborhood sampling has been de-facto in training GNN. The issue of changing graph structure in GCL has been documented and studied in [2].

[1]  "Cglb: Benchmark tasks for continual graph learning." Advances in Neural Information Processing Systems 35 (2022): 13006-13021.

[2] "Towards robust graph incremental learning on evolving graphs." International Conference on Machine Learning. PMLR, 2023.

**Questions:**

see weakness

---

> ### Author Response · Authors · 2024-11-23
>
> We appreciate your thoughtful review and the detailed feedback on the paper. Below, we address each identified weakness while clarifying our paper's contributions.
>
> 1. - a. The formulation of the OCGL framework is provided in section 3 of the paper. While we acknowledge that the formalism is limited, in section 3.1 the growing network is formally defined. Specific definitions of the learner are not provided as to accommodate multiple solutions, and the requirements for efficiency are explored in the considerations on mini-batching and neighborhood expansion. The objective we wanted to achieve with our framework was to maintain a general enough setting that can be applied in multiple scenarios, therefore a stringent formalism would have harmed the scope of the OCGL framework.
>     - b. We acknowledge that a detailed comparison between CGL and OCGL is not provided, yet they are compared at the end of section 2 and in section 3.1: the key difference between the two is that in CGL the streams consist of graph snapshots (generally subgraphs induced by the current task) on which models are trained offline with multiple passes (possibly also with batching), while in OCGL the stream consists of small mini-batches of nodes, which are not used for training after a new batch arrives.
> 2. We respectfully disagree with the reviewer's statement that continual learning is an inherently online setting: while we agree on the points on continual learning and that even for online systems storing historical data is also allowed, there is a fundamental difference between standard continual learning and online continual learning. While in standard continual learning training on each task is performed offline, with multiple passes over the data until convergence, the requirements of online learning involve a single pass over the data. Such a distinction is clearly identified in the literature [1,2]. We would thus kindly invite the reviewer to point out the specific statements in our manuscript which are deemed factually incorrect, so that we may correct them.
>     Regarding the task boundary, we agree that there have been studies that looked at online continual learning without them, yet this is missing in the CGL literature, as even the referenced paper only proposes it as future work, and the same authors identify it as an open research direction in their recent survey [3]. This further highlights the contribution of our work. As to the definition of task boundary, a clarifying sentence has been introduced in the manuscript.
> 3. We wish to clarify that the objective of our paper is not to propose novel techniques for continual learning on graphs, but rather our primary contribution is the introduction of the Online Continual Graph Learning setting, which has distinct peculiarities compared to standard CGL.
>     This contribution holds particular significance because, although certain studies in the CGL literature describe their frameworks as streaming, they often lack direct comparability with the OCL domain. Moreover, a clear and consistent framework like the one we introduce is absent. By establishing a well-defined problem space, identifying potential challenges, and providing a comprehensive benchmark, our work serves as a foundation for creating and evaluating new methods for efficient online continual graph learning.
>     While certainly the issue of changing graph structure has been documented in multiple surveys, we believe that it poses specific challenges associated with the online setting, in particular regarding mini-batching and neighborhood expansion. Similarly for neighborhood expansion and sampling, while this issue is not unique to our setting, here sampling is not simply a matter of efficiency, but a requirement of the online setting. We want to stress how our contribution does not lie in the introduction of new techniques (as we point out in section 3.2 that sampling is a simple solution for neighborhood expansion problem for scaling GNNs to large graphs), but the introduction of a previously ignored setting, alongside a consideration of the issue it presents and that need to be further addressed in future studies.
>
> [1] Zheda Mai, Ruiwen Li, Jihwan Jeong, David Quispe, Hyunwoo Kim, and Scott Sanner, "Online continual learning in image classification: An empirical survey", Neurocomputing, 2022
>
> [2] Albin Soutif–Cormerais, Antonio Carta, Andrea Cossu, Julio Hurtado, Vincenzo Lomonaco, Joost Van De Weijer, and Hamed Hemati, "A Comprehensive Empirical Evaluation on Online Continual Learning", in 2023 IEEE/CVF International Conference on Computer Vision Workshops (ICCVW), 2023
>
> [3] Xikun Zhang, Dongjin Song, and Dacheng Tao, "Continual Learning on Graphs: Challenges, Solutions, and Opportunities", arXiv:2402.11565, 2024

---

> > ### Comment · Reviewer_n8Q1 · 2024-11-25
> >
> > I thank the author for the diligent response. However, I still have not got clear answers to my concerns. For example, the new response persists the main contribution of the paper is the study of a new setting. I still fail to see what is the new setting, how it is different from the typical graph continual learning, and why this setting is important.
> >
> > As such, I will maintain the original score.

---

> > > ### Author Response · Authors · 2024-11-25
> > >
> > > We thank the reviewer for their feedback and the opportunity to clarify the Online Continual Graph Learning (OCGL) setting. OCGL addresses a gap between the literature on Online Continual Learning (OCL) and Continual Graph Learning (CGL) by introducing a learning paradigm tailored to real-time graph data. In the camera-ready version of the manuscript, we will introduce a paragraph to specifically compare the OCGL with CGL. As mentioned in the problem formulation of section 3.1, the distinctions are as follows:
> > > - Data stream and granularity: in CGL, the data stream typically consists of graph snapshots (e.g., subgraphs or tasks) available for offline training with multiple passes. In contrast, OCGL processes data as individual nodes or small mini-batches, requiring immediate updates and supporting only single-pass learning.
> > > - Training methodology: standard CGL assumes offline settings with relaxed computational constraints. OCGL enforces stricter online constraints, limiting computational overhead and memory usage, and specifically requires bounded processing time for each mini-batch. This allows the model to adapt quickly to distribution changes, allowing to make anytime predictions.
> > > - Task boundaries: CGL frequently relies on explicit task boundaries for training and evaluation, while OCGL operates in a task-free or boundary-agnostic manner. This makes OCGL suitable for dynamic, evolving graphs where distribution shifts occur without clear demarcations.
> > >
> > > The OCGL framework thus addresses significant gaps in the existing literature by adapting online continual learning principles to dynamic graph-structured data, especially the computational efficiency which leads to our proposal of neighborhood sampling. This framework formalizes a novel setting that combines the challenges of evolving graph data with the stringent constraints of online learning, such as single-pass training, task-free learning, and limited memory and computational resources. By doing so, OCGL captures the demands of real-world scenarios, such as social networks or recommender systems, where data arrives incrementally, and predictions must be made in real time.
> > >
> > > Overall, the contributions of our work are twofold: establishing OCGL as a well-defined and practical problem space and creating the foundation for future research by identifying key challenges and providing the tools to address them.

---

> > > > ### Comment · Reviewer_n8Q1 · 2024-11-29
> > > >
> > > > Thank you for the further response.
> > > > However, I still do not find sufficient/direct answers to my questions, as optimizing for memory/computation resources are also desired/standard objectives for CGL methods.
> > > >
> > > > I will suggest that in the future version of the paper,  the author
> > > >
> > > > 1. provide a formal side-by-side comparison with a quantitative description rather than a qualitative description between CGL and OCL  (as this is one of the claimed main contributions)
> > > >
> > > > 2. provide a concrete toy example of the proposed setting, illustrate how the task boundary changes different from CGL and explain why the existing CGL methods are not applicable/effective in this setting.

---

### Official Review · Reviewer_gFtT · 2024-10-30

**Soundness:** 2
**Presentation:** 2
**Contribution:** 2
**Rating:** 5
**Confidence:** 4

**Summary:**

The paper introduces the Online Continual Graph Learning (OCGL) framework to handle non-stationary streaming data in graph structures. It benchmarks several continual learning methods on four datasets, adapting them for the online graph learning scenario. The authors propose a neighborhood sampling strategy to address the issue of neighborhood expansion in Graph Neural Networks (GNNs) and reduce computational complexity.

**Strengths:**

1. The introduction of the Online Continual Graph Learning (OCGL) framework extends continual learning to dynamic, graph-structured data.

2. The paper provides a thorough evaluation of multiple continual learning methods, adapting them for online graph learning.

3. The proposed neighborhood sampling strategy effectively addresses the computational and memory challenges of multi-hop neighborhood aggregation in GNNs.

**Weaknesses:**

1. The benchmarks focus mainly on node classification tasks, and extending the framework to more diverse graph-based applications (e.g., edge prediction, link prediction) could strengthen the paper's contributions.

2. The paper primarily compares traditional continual learning methods adapted for the Online Continual Graph Learning (OCGL) framework. It does not include comparisons with more recent state-of-the-art continual graph learning methods proposed in the recent three years, such as MSCGL[1] and UGCL[2].

    [1] J. Cai, X. Wang, C. Guan, Y. Tang, J. Xu, B. Zhong, and W. Zhu, ''Multimodal continual graph learning with neural architecture search,'' in Proceedings of the ACM Web Conference, 2022, pp.1292–1300.

    [2] T. D. Hoang, D. V. Tung, D.-H. Nguyen, B.-S. Nguyen, H. H.Nguyen, and H. Le, ''Universal graph continual learning,'' Transactions on Machine Learning Research, 2023.

3. While the sampling strategy improves computational efficiency, it can negatively impact model accuracy.

4. The paper predominantly concentrates on experimental evaluation and lacks an in-depth theoretical analysis of the proposed method's properties, such as convergence, computational complexity, and theoretical bounds on forgetting.

**Questions:**

See weaknesses.

---

> ### Author Response · Authors · 2024-11-23
>
> We appreciate your thoughtful review and valuable feedback. Below, we provide point-by-point responses to address the weaknesses and questions raised while clarifying our paper's contributions and setting.
>
> 1. Thank you for this suggestion. While our primary focus was on node classification due to its prominence in graph learning literature, we agree that extending the framework to tasks like edge prediction would significantly broaden its applicability. We will revise the conclusion to explicitly outline these extensions as important steps for future research.
> 2. Thank you for suggesting the inclusion of more recent methods for continual graph learning. While we initially focused on widely recognized baselines to ensure a robust comparison within the established continual learning literature, we agree that evaluating newer methods is essential to accurately reflect state-of-the-art performance. We thank you for the pointers to recent literature. However, most recent methods in the CGL literature violate the constraints of the OCGL setting either due to expensive update steps between tasks (such as a costly neural architecture search) or due to inefficient use of past data. As an example, the referenced "Universal graph continual learning" proposed a global structure distillation which requires to compute node embeddings for the entire graph, and a local structure distillation which still requires to compute the full embedding of buffer nodes and all their neighbors. Most of these methods therefore are not applicable in our setting. We have nonetheless identified in "Sparsified Subgraph Memory" [1] a suitable additional baseline, for which we have already launched experiments, whose results we will include in the final version of the paper.
> 3. The proposed simple solution of sampling to address the issue of neighborhood expansion is only a first step in tackling the issue. As we observe at the end of section 7, the (expected) negative impact on model accuracy indicates that further research is required to fully address the issue. Rather than a contribution by itself, the proposal of the sampling method serves to bring the attention to the specific issue of neighborhood expansion, which requires more consideration in this setting.
> 4. We appreciate this observation and agree that a sound theoretical analysis of convergence and bounds on forgetting adds value to continual learning papers. Nonetheless, unfortunately such an analysis is not straightforward for most methods, and is not provided even in the papers that introduce them. Such a detailed model analysis is therefore out of the scope of our work, which introduces a new, challenging setting for continual learning on graphs, rather that proposing a new method, or proving theoretical properties of existing ones.
>
> [1] Xikun Zhang, Dongjin Song, and Dacheng Tao, "Sparsified Subgraph Memory for Continual Graph Representation Learning", in 2022 IEEE International Conference on Data Mining (ICDM), 2022

---

> ### Comment · Reviewer_gFtT · 2024-11-25
>
> I appreciate the authors for their efforts and responses. The OCGL framework and benchmark represent valuable foundational steps for exploring online continual graph learning. However, as a foundational framework, the evaluation should be more comprehensive, encompassing diverse graph-based tasks beyond node classification, such as link prediction or graph classification, to demonstrate the framework’s broader applicability. Additionally, the current work relies on a limited set of datasets, which restricts the generalizability of the proposed benchmark.
>
> Given these limitations and the lack of immediate impact demonstrated by the framework, the contribution appears less significant compared to other works at this conference. Thus, I will maintain my original assessment.

---

### Official Review · Reviewer_zdzj · 2024-11-02

**Soundness:** 2
**Presentation:** 2
**Contribution:** 2
**Rating:** 6
**Confidence:** 4

**Summary:**

This paper aims to formulate the setting of online continual graph learning considering the efficiency of batch processing and graph topology, and proposes a set of benchmark datasets for online continual graph learning. Additionally, from the technical perspective, the authors address the challenge of GNN memory usage.

Within the context of online continual graph learning, the graphs are defined as a time series, in which the graph snapshot at each time stamp t contains the nodes and edges collected from the starting time till t. Each new snapshot is created when a new node is added. The newly attached information includes the new node, its neighbors, and the node features.

**Strengths:**

1. Online continual graph learning has not been fully explored, and this work makes some contribution in this direction.

2. Compared to existing continual graph learning works, this work adopts a more practical hyperparameter selection strategy that only use a few tasks.

**Weaknesses:**

1. The main weakness is the inconsistence between the proposed setting and the actual experiments. Although the paper described an online learning setting, but the task construction in experiments is still same as the continual graph learning setting with task boundaries. As mentioned in the paper, 'the graph grow with nodes from two new classes at a time', then the incremental manner is same as a normal class incremental learning instead of an online learning setting. I would recommend that the experiments should be consistent with the proposed setting,in which each new snapshot could contain one node or a mini-batch of new nodes, but not necessarily a new task containing new classes.

2. The adopted baselines are a little bit out of date. Besides, only TWP is specially designed for graph data, while the others don't consider the graph structures. Admittedly the authors have discussed why some baselines are not adopted, but the mentioned ones are all proposed no later than 2021. Therefore, it is not convincing enough that the adopted methods can represent the state-of-the-art performance. I would recommend that the recent continual graph learning works proposed from 2022 to 2024 could be thoroughly investigated, discussed, and compared whenever appropriate.

**Questions:**

1. In the Mini-batching part of Section 3.1, what does it mean by 'L>1 is not in contrast with the growing mechanism of the graph'? I could understand what the authors want to express should be that the entire graph may be required for aggregating multi-hop information, but the writing here seems confusing.

2. It is a little bit confusing whether the proposed strategy allows the model to access the entire graph that contains previous nodes. It is stated that the up-to-date graph Gt is stored in a Past Information Store (PIS) system, but only allow limited use of information from PIS. It is unclear what kind of usage is deemed as 'limited'. Additionally, it is also stated that the PIS is different from a 'eventual memory buffer'. This is also confusing. If the PIS contains the completely graph with all previous information, then what is the role of the 'eventual memory buffer', and why we still need such a buffer?

3. In the 'training details' part in the experiment section, when talking about the batch size, how is each batch used? Given a new task with N data, will the model be trained on the N data for several epochs, and in each epoch, the batches are fed into the model sequentially?

---

> ### Author Response · Authors · 2024-11-23
>
> We thank you for the review. Below, we respond to the identified weaknesses and questions while addressing the concerns that were raised.
>
> Weaknesses:
> 1. We believe that the first and main weakness identified by the reviewer stems from a misunderstanding of our experimental setting. We rewrote a sentence to make it clearer. Specifically, with "the graph will gradually grow with nodes from two new classes at a time" we mean that we stream mini-batches of nodes from two classes at a time.  In the conventional incremental setting, the data (nodes) of a single task (nodes of two classes) would be available all at once, allowing for multiple passes until convergence. On the contrary, in our experiments, for each pair of classes multiple small mini-batches are processed one by one in an online fashion (without revisiting them later), before passing to the next pair of classes. Therefore our experiments follow exactly the proposed setting, with a class-incremental node stream definition.
> 2. We appreciate the suggestion to incorporate more recent methods for continual graph learning. However, please note that the most recent methods in the CGL literature violate the constraints of the OCGL setting either due to expensive update steps between tasks or due to inefficient use of past data. Most of these methods therefore are not applicable in our setting. While we selected widely recognized baselines to provide a robust comparison within the continual learning literature, we agree that newer methods should be evaluated to reflect state-of-the-art performance. For this purpose, in the final version of the paper we will include results for "Sparsified Subgraph Memory" [1], for which we have already launched the experiments.
>
> Questions:
> 1. We thank you for bringing up this point of confusion, which certainly requires a rewording. We intended to emphasize that aggregating multi-hop information does not fundamentally conflict with the evolving nature of the graph. Yet, this necessitates storage of past information (see next point) and raises the problem of neighborhood expansion.
> 2. The purpose of a Past Information Storage is to have access to multiple hops of neighboring nodes for message passing when new ones arrive. In this context, the "limited" usage is defined in the preceding sentence with the requirement of bounded complexity for processing each mini-batch, to ensure efficiency in the presence of the neighborhood expansion issue. The role of the PIS is thus distinct from the memory buffer of replay methods: it serves as a "database" where even a huge graph could be stored, and its use is similar to what is done in traditional CGL when accessing inter-task edges. In such CGL settings, while not explicitly stated, it is de-facto used nonetheless as multiple hops of past nodes are required. An alternative, equivalent formulation would be to require that each node arrives equipped also with its l-hop neighborhood, or a subset of it. The memory buffer instead is a more limited storage of samples that is not used for the construction of neighboring information of mini-batch nodes, but it is used by the continual learning method to preventing forgetting.
> 3. As clarified in response to the first identified weakness, given a new task with N nodes, the model will be trained on N/batch\_size mini-batches in an online fashion. We consider multiple passes only on each mini-batch before passing to the next.
>
> [1] Xikun Zhang, Dongjin Song, and Dacheng Tao, "Sparsified Subgraph Memory for Continual Graph Representation Learning", in 2022 IEEE International Conference on Data Mining (ICDM), 2022

---

> > ### Comment · Reviewer_zdzj · 2024-11-24
> >
> > Thanks for the detailed response from the authors.
> >
> > Previously, my major concern is on the online learning setting. After reading the explanation from the authors, I understand that it is indeed the online setting proposed at the beginning of the paper. Therefore I would like to increase my rating to 6.
> >
> > Additionally, I would also encourage the authors to clarify the relationship between PIS and memory buffer. Although PIS is used to access the neighborhood while memory buffer is used to prevent forgetting, the information retained by these two mechanisms are same. i.e. PIS already contains all information of the memory buffer, and it seems that the memory buffer do not store additional information but could index and retrieve from the PIS.

---

> > > ### Author Response · Authors · 2024-11-25
> > >
> > > We greatly appreciate the reviewer’s follow-up. While it is true that the memory buffer is a subset of the information contained in the PIS, their difference is computational. While the PIS is allowed to grow to accommodate the graph stream, the memory buffer has to maintain constant dimension to be used by continual learning strategies to prevent forgetting. In some cases in practice the memory buffer can be implemented by a simple indexing and retrieval from the PIS, especially if the graph is small enough to fit in memory. Instead, if the graph becomes huge, which is a case we want to be prepared for in the OCGL setting, this becomes impossible, as retrieval from the PIS becomes less efficient, as it could be stored for example in a distributed database. Finally, there may be cases in which we only have a "virtual" PIS, where we do not have a storage of the graph but we may explore it when obtaining mini-batches (or if new nodes come with neighborhood information). Thus we believe it is best to keep the PIS and the memory buffer conceptually separate in our framework.

---

### Official Review · Reviewer_RugN · 2024-11-05

**Soundness:** 3
**Presentation:** 3
**Contribution:** 2
**Rating:** 6
**Confidence:** 4

**Summary:**

The paper introduces a novel framework, OCGL, which addresses the challenges of applying continual learning principles to graph-structured data in an online setting. It innovatively formulates the problem of learning from a continuously evolving graph, emphasizing the necessity to manage computational complexity and avoid catastrophic forgetting—a common issue where a model loses previously learned information upon learning new data. To facilitate research in this area, the authors develop a benchmarking environment with tailored datasets to evaluate various continual learning methods adapted to graph settings. They also propose practical solutions such as neighborhood sampling to maintain computational efficiency as the graph grows.

**Strengths:**

1. One of the paper's main strengths is the formal introduction of the Online Continual Graph Learning (OCGL) framework.
2. The authors develop a benchmarking environment specifically for OCGL, including multiple datasets and evaluations of various continual learning methods.
3. The experimental setup and the detailed analysis provided in the paper are thorough and well-constructed.

**Weaknesses:**

1. The proposed problem is novel, however, the detailed appliable scenario for such OCGL framework should be further explained, especially on graph data.
2. The baselines chosen in this paper are all Continual learning methods. More methods for the online learning setting should be included.
3. Also, as a benchmark paper, it would be beneficial to introduce more new datasets.
4. The contribution of this paper seems limited to me. The authors introduced a new problem setting OCGL for graph learning and presented a benchmarking environment for OCGL, but did not propose a novel method to solve this problem. Although I understand benchmarking papers are also important to the research community, I believe that the contribution in this case may not be significantly sufficient for inclusion in this conference.
5. The third contribution, using random sampling to address the complexity of multi-hop aggregation, is a very easy-to-get idea, and seems trivial to me.

**Questions:**

see above.

---

> ### Author Response · Authors · 2024-11-23
>
> We thank the reviewer for the detailed comments. Here are point-by-point answers to the identified weaknesses:
> 1. We acknowledge the importance of elaborating on the scenarios where OCGL can be impactful. Specifically, OCGL can be used in all the applications of graph continual learning where quick model adaptation is required for timely predictions after distribution shifts. To address this we expanded the beginning of section 3 on the introduction of the OCGL framework for streaming graph data.
> 2. While all the baselines for the paper are Continual Learning methods, we note how A-GEM and MAS are in fact  "onlinable" by design, as the learning protocol of A-GEM involves a single pass over the training stream and the parameter importance of MAS is updated in an online fashion. While additional baselines could be considered, we believe that the current ones are a representative selection of Continual Learning strategies. As pointed out also by reviewers zdzj and gFtT, we would rather expand our selection of baselines with a more recent GCL methods. In particular, in the final version of the paper we will include results for "Sparsified Subgraph Memory" [1], for which we have alredy launched the experiments.
> 3. We recognize the value of diversity in benchmarking datasets. While our study already spans four well-established graph datasets (CoraFull, Arxiv, Reddit, and Amazon Computer), we plan to expand this suite in future work to include additional graph datasets. The number of considered dataset is nonetheless in line with similar literature in GCL and OCL [2,3,4].
> 4. We wish to clarify that the primary aim of this work is to formalize the OCGL framework and provide a foundational benchmark. We believe this is an essential first step for further exploration and innovation in this field. By defining the problem space, analyzing possible problems and offering a robust benchmarking environment, our paper lays the groundwork for developing and testing novel algorithms. We believe that this contribution is particularly relevant as in the CGL literature there are some studies that refer to their setting as streaming, yet they are arguably not comparable to the OCL literature, and there is no well defined setting such as the one we propose with our work.
> 5. While random sampling might appear straightforward, we adopted this as a first, pragmatic solution to highlight the neighborhood expansion problem and to demonstrate its impact. We acknowledge this limitation and propose in the conclusion that future research should develop more sophisticated strategies tailored to OCGL. In addition, we believe that issue is in need to be raised not only in the online setting, but also in standard CGL, where some methods implicitly use past data thanks to multiple hops of message passing into past task nodes. We hope therefore that raising the issue can lead to more careful and efficient design strategies for GCL.
>
> [1] Xikun Zhang, Dongjin Song, and Dacheng Tao, "Sparsified Subgraph Memory for Continual Graph
> Representation Learning", in 2022 IEEE International Conference on Data Mining (ICDM), 2022
>
> [2] Xikun Zhang, Dongjin Song, and Dacheng Tao, "CGLB: Benchmark Tasks for Continual Graph
> Learning", Advances in Neural Information Processing Systems, 2022
>
> [3] Zheda Mai, Ruiwen Li, Jihwan Jeong, David Quispe, Hyunwoo Kim, and Scott Sanner, "Online
> continual learning in image classification: An empirical survey", Neurocomputing, 2022
>
> [4] Albin Soutif–Cormerais, Antonio Carta, Andrea Cossu, Julio Hurtado, Vincenzo Lomonaco, Joost
> Van De Weijer, and Hamed Hemati, "A Comprehensive Empirical Evaluation on Online Continual
> Learning", in 2023 IEEE/CVF International Conference on Computer Vision Workshops (ICCVW), 2023

---

> ### Comment · Reviewer_RugN · 2024-11-25
>
> I thank the author for the response, which has resolved some of my concerns. However, I still think the technical and experimental contributions are somewhat insufficient, so I will maintain my score.

---

### Meta-Review · Area_Chair_CD5L · 2024-12-20

**Metareview:**

This paper aims to address the challenges of applying continual learning principles to graph-structured data in an online setting. Reviewers agreed that this paper presents an interesting framework for online continual graph learning (OCGL) and also proposes novel ideas such as a neighborhood sampling strategy. However, reviewers raised concerns about insufficient justifications of the problem setting, insufficient baselines and datasets, the lack of theoretical analysis, etc. Although some of the concerns have been addressed during the rebuttal and discussion stages, reviewers and the AC still found the paper in its current version is not ready for publication at ICLR yet.

**Additional Comments On Reviewer Discussion:**

Reviewers raised concerns about insufficient justifications of the problem setting, insufficient baselines and datasets, the lack of theoretical analysis, etc. The detailed responses have addressed some of these concerns. However, during the post-rebuttal discussions, reviewers still had some concerns that cannot be resolved by a minor revision. For instance, the unique technical challenges for the proposed problem setting remain unclear. Also, the evaluation only covers a limited set of datasets, which restricts the generalizability of the proposed benchmark.

---

### Decision · Program_Chairs · 2025-01-22

Reject